# Associating Objects with Transformers for Video Object Segmentation

**Zongxin Yang**[1,2], **Yunchao Wei**[3,4], **Yi Yang**[1]

[1] CCAI, College of Computer Science and Technology, Zhejiang University  [2] Baidu Research
[3] Institute of Information Science, Beijing Jiaotong University
[4] Beijing Key Laboratory of Advanced Information Science and Network
{zongxinyang1996, wychao1987, yee.i.yang}@gmail.com

## Abstract

This paper investigates how to realize better and more efficient embedding learning to tackle the semi-supervised video object segmentation under challenging multi-object scenarios. The state-of-the-art methods learn to decode features with a single positive object and thus have to match and segment each target separately under multi-object scenarios, consuming multiple times computing resources. To solve the problem, we propose an Associating Objects with Transformers (AOT) approach to match and decode multiple objects uniformly. In detail, AOT employs an identification mechanism to associate multiple targets into the same high-dimensional embedding space. Thus, we can simultaneously process multiple objects' matching and segmentation decoding as efficiently as processing a single object. For sufficiently modeling multi-object association, a Long Short-Term Transformer is designed for constructing hierarchical matching and propagation. We conduct extensive experiments on both multi-object and single-object benchmarks to examine AOT variant networks with different complexities. Particularly, our R50-AOT-L outperforms all the state-of-the-art competitors on three popular benchmarks, *i.e.*, YouTube-VOS (84.1% $\mathcal{J}\&\mathcal{F}$), DAVIS 2017 (84.9%), and DAVIS 2016 (91.1%), while keeping more than $3\times$ faster multi-object run-time. Meanwhile, our AOT-T can maintain real-time multi-object speed on the above benchmarks. Based on AOT, we ranked $\mathbf{1^{st}}$ in the 3rd Large-scale VOS Challenge.

## 1 Introduction

Video Object Segmentation (VOS) is a fundamental task in video understanding with many potential applications, including augmented reality [25] and self-driving cars [52]. The goal of semi-supervised VOS, the main task in this paper, is to track and segment object(s) across an entire video sequence based on the object mask(s) given at the first frame.

Thanks to the recent advance of deep neural networks, many deep learning based VOS algorithms have been proposed recently and achieved promising performance. STM [26] and its following works [34, 23] leverage a memory network to store and read the target features of predicted past frames and apply a non-local attention mechanism to match the target in the current frame. FEELVOS [41] and CFBI [50, 51] utilize global and local matching mechanisms to match target pixels or patches from both the first and the previous frames to the current frame.

Even though the above methods have achieved significant progress, the above methods learn to decode scene features that contain a single positive object. Thus under a multi-object scenario, they have to match each object independently and ensemble all the single-object predictions into a multi-object segmentation, as shown in Fig. 1a. Such a post-ensemble manner eases network architectures' design

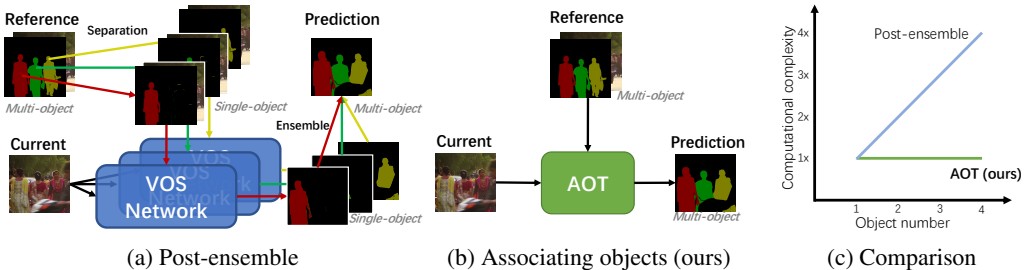

(a) Post-ensemble      (b) Associating objects (ours)      (c) Comparison

Figure 1: VOS methods (*e.g.*, [50, 34]) process multi-object scenarios in a post-ensemble manner (a). In contrast, our AOT associates all the objects uniformly (b), leading to better efficiency (c).

since the networks are not required to adapt the parameters or structures for different object numbers. However, modeling multiple objects independently, instead of uniformly, is inefficient in exploring multi-object contextual information to learn a more robust feature representation for VOS. In addition, processing multiple objects separately yet in parallel requires multiple times the amount of GPU memory and computation for processing a single object. This problem restricts the training and application of VOS under multi-object scenarios, especially when computing resources are limited.

To solve the problem, Fig. 1b demonstrates a feasible approach to associate and decode multiple objects uniformly in an end-to-end framework. Hence, we propose an Associating Objects with Transformers (AOT) approach to match and decode multiple targets uniformly. First, an identification mechanism is proposed to assign each target a unique identity and embed multiple targets into the same feature space. Hence, the network can learn the association or correlation among all the targets. Moreover, the multi-object segmentation can be directly decoded by utilizing assigned identity information. Second, a Long Short-Term Transformer (LSTT) is designed for constructing hierarchical object matching and propagation. Each LSTT block utilizes a long-term attention for matching with the first frame's embedding and a short-term attention for matching with several nearby frames' embeddings. Compared to the methods [26, 34] utilizing only one attention layer, we found hierarchical attention structures are more effective in associating multiple objects.

We conduct extensive experiments on two popular multi-object benchmarks for VOS, *i.e.*, YouTube-VOS [48] and DAVIS 2017 [31], to validate the effectiveness and efficiency of the proposed AOT. Even using the light-weight Mobilenet-V2 [33] as the backbone encoder, the AOT variant networks achieve superior performance on the validation 2018 & 2019 splits of the large-scale YouTube-VOS (ours, $\mathcal{J}\&\mathcal{F}$ **82.6**∼ **84.5**% & **82.2**∼ **84.5**%) while keeping more than **2**× faster multi-object run-time (**27.1**∼ **9.3**FPS) compared to the state-of-the-art competitors (*e.g.*, CFBI [50], 81.4% & 81.0%, 3.4FPS). We also achieve new state-of-the-art performance on both the DAVIS-2017 validation (**85.4**%) and testing (**81.2**%) splits. Moreover, AOT is effective under single-object scenarios as well and outperforms previous methods on DAVIS 2016 [30] (**92.0**%), a popular single-object benchmark. Besides, our smallest variant, AOT-T, can maintain real-time multi-object speed on all above benchmarks (**51.4**FPS on 480p videos). Particularly, AOT ranked **1**st in the Track 1 (Video Object Segmentation) of the 3rd Large-scale Video Object Segmentation Challenge.

Overall, our contributions are summarized as follows:

- We propose an identification mechanism to associate and decode multiple targets uniformly for VOS. For the first time, multi-object training and inference can be efficient as single-object ones, as demonstrated in Fig. 1c.
- Based on the identification mechanism, we design a new efficient VOS framework, *i.e.*, Long Short-Term Transformer (LSTT), for constructing hierarchical multi-object matching and propagation. LSTT achieves superior performance on VOS benchmarks [48, 31, 30] while maintaining better efficiency than previous state-of-the-art methods. To the best of our knowledge, LSTT is the first hierarchical framework for object matching and propagation by applying transformers [39] to VOS.

## 2 Related Work

**Semi-supervised Video Object Segmentation.** Given one or more annotated frames (the first frame in general), semi-supervised VOS methods propagate the manual labeling to the entire video

sequence. Traditional methods often solve an optimization problem with an energy defined over a graph structure [4, 40, 2]. In recent years, VOS methods have been mainly developed based on deep neural networks (DNN), leading to better results.

Early DNN methods rely on fine-tuning the networks at test time to make segmentation networks focus on a specific object. Among them, OSVOS [7] and MoNet [47] fine-tune pre-trained networks on the first-frame ground-truth at test time. OnAVOS [42] extends the first-frame fine-tuning by introducing an online adaptation mechanism. Following these approaches, MaskTrack [29] and PReM [24] utilize optical flow to help propagate the segmentation mask from one frame to the next. Despite achieving promising results, the test-time fine-tuning restricts the network efficiency.

Recent works aim to achieve a better run-time and avoid using online fine-tuning. OSMN [49] employs one convolutional network to extract object embedding and another one to guide segmentation predictions. PML [9] learns pixel-wise embedding with a nearest neighbor classifier, and VideoMatch [18] uses a soft matching layer that maps the pixels of the current frame to the first frame in a learned embedding space. Following PML and VideoMatch, FEELVOS [41] and CFBI [50, 51] extend the pixel-level matching mechanism by additionally matching between the current frame and the previous frame. RGMP [46] also gathers guidance information from both the first frame and the previous frame but uses a siamese encoder with two shared streams. STM [26] and its following works (*e.g.*, EGMN [23] and KMN [34]) leverage a memory network to embed past-frame predictions into memory and apply a non-local attention mechanism on the memory to decode the segmentation of the current frame. SST [13] utilizes attention mechanisms in a different way, *i.e.*, transformer blocks [39] are used to extract pixel-level affinity maps and spatial-temporal features. The features are target-agnostic, instead of target-aware like our LSTT, since the mask information in past frames is not propagated and aggregated in the blocks. Instead of using matching mechanisms, LWL [6] proposes to use an online few-shot learner to learn to decode object segmentation.

The above methods learn to decode features with a single positive object and thus have to match and segment each target separately under multi-object scenarios, consuming multiple times computing resources of single-object cases. The problem restricts the application and development of the VOS with multiple targets. Hence, we propose our AOT to associate and decode multiple targets uniformly and simultaneously, as efficiently as processing a single object.

**Visual Transformers.** Transformers [39] was proposed to build hierarchical attention-based networks for machine translation. Similar to Non-local Neural Networks [43], transformer blocks compute correlation with all the input elements and aggregate their information by using attention mechanisms [5]. Compared to RNNs, transformer networks model global correlation or attention in parallel, leading to better memory efficiency, and thus have been widely used in natural language processing (NLP) tasks [11, 32, 37]. Recently, transformer blocks were introduced to many computer vision tasks, such as image classification [12, 38, 22], object detection [8]/segmentation [44], and image generation [27], and have shown promising performance compared to CNN-based networks.

Many VOS methods [19, 26, 23, 34] have utilized attention mechanisms to match the object features and propagate the segmentation mask from past frames to the current frames. Nevertheless, these methods consider only one positive target in the attention processes, and how to build hierarchical attention-based propagation has been rarely studied. In this paper, we carefully design a long short-term transformer block, which can effectively construct multi-object matching and propagation within hierarchical structures for VOS.

## 3 Revisit Previous Solutions for Video Object Segmentation

In VOS, many common video scenarios have multiple targets or objects required for tracking and segmenting. Benefit from deep networks, current state-of-the-art VOS methods [26, 50] have achieved promising performance. Nevertheless, these methods focus on matching and decoding a single object. Under a multi-object scenario, they thus have to match each object independently and ensemble all the single-object predictions into a multi-object prediction, as demonstrated in Fig. 1a. Let $F^{\mathcal{N}}$ denotes a VOS network for predicting single-object segmentation, and $A$ is an ensemble function such as $softmax$ or the soft aggregation [26], the formula of such a post-ensemble manner for processing $N$ objects is like,

$$Y' = A(F^{\mathcal{N}}(I^t, I^{\mathbf{m}}, Y_1^{\mathbf{m}}), ..., F^{\mathcal{N}}(I^t, I^{\mathbf{m}}, Y_N^{\mathbf{m}})), \tag{1}$$

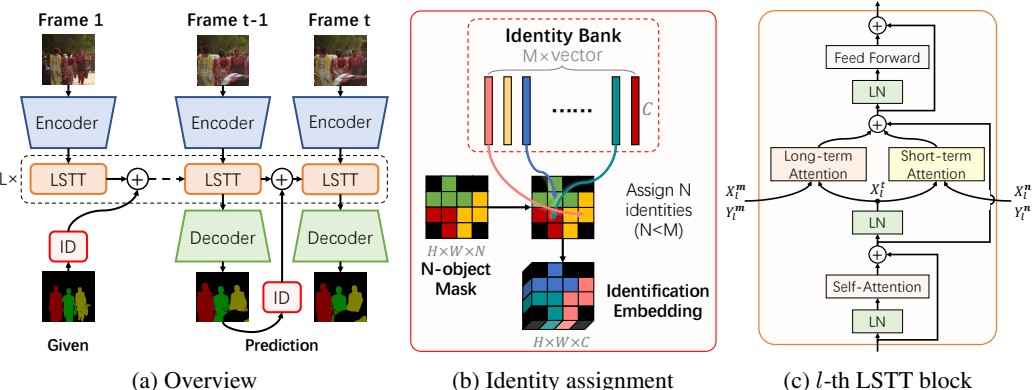

| (a) Overview | (b) Identity assignment | (c) $l$-th LSTT block |

Figure 2: (a) The overview of our Associating Objects with Transformers (AOT). The multi-object masks are embedded by using our Identification mechanism. Moreover, a $L$-layer Long Short-Term Transformer is responsible for matching multiple objects uniformly and hierarchically. (b) An illustration of the IDentity assignment (ID) designed for transferring a $N$-object mask into an identification embedding. (c) The structure of an LSTT block. LN: layer normalization [3].

where $I^t$ and $I^{\mathbf{m}}$ denote the image of the current frame and memory frames respectively, and $\{Y_1^{\mathbf{m}}, ..., Y_N^{\mathbf{m}}\}$ are the memory masks (containing the given reference mask and past predicted masks) of all the $N$ objects. This manner extends networks designed for single-object VOS into multi-object applications, so there is no need to adapt the network for different object numbers.

Although the above post-ensemble manner is prevalent and straightforward in the VOS field, processing multiple objects separately yet in parallel requires multiple times the amount of GPU memory and computation for matching a single object and decoding the segmentation. This problem restricts the training and application of VOS under multi-object scenarios when computing resources are limited. To make the multi-object training and inference as efficient as single-object ones, an expected solution should be capable of associating and decoding multiple objects uniformly instead of individually. To achieve such an objective, we propose an identification mechanism to embed the masks of any number (required to be smaller than a pre-defined large number) of targets into the same high-dimensional space. Based on the identification mechanism, a novel and efficient framework, *i.e.*, Associating Objects with Transformers (AOT), is designed for propagating all the object embeddings uniformly and hierarchically, from memory frames to the current frame.

As shown in Fig. 1b, our AOT associates and segments multiple objects within an end-to-end framework. For the first time, processing multiple objects can be as efficient as processing a single object (Fig. 1c). Compared to previous methods, our training under multi-object scenarios is also more efficient since AOT can associate multiple object regions and learn contrastive feature embeddings among them uniformly.

## 4 Associating Objects with Transformers

In this section, we introduce our identification mechanism proposed for efficient multi-object VOS. Then, we design a new VOS framework, *i.e.*, long short-term transformer, based on the identification mechanism for constructing hierarchical multi-object matching and propagation.

### 4.1 Identification Mechanism for Multi-object Association

Many recent VOS methods [26, 23, 34] utilized attention mechanisms and achieved promising results. To formulate, we define $Q \in \mathbb{R}^{HW \times C}$, $K \in \mathbb{R}^{THW \times C}$, and $V \in \mathbb{R}^{THW \times C}$ as the query embedding of the current frame, the key embedding of the memory frames, and the value embedding of the memory frames respectively, where $T$, $H$, $W$, $C$ denote the temporal, height, width, and channel dimensions. The formula of a common attention-based matching and propagation is,

$$Att(Q, K, V) = Corr(Q, K)V = softmax(\frac{QK^{tr}}{\sqrt{C}})V, \tag{2}$$

where a matching map is calculated by the correlation function $Corr$, and then the value embedding, $V$, will be propagated into each location of the current frame.

In the common single-object propagation [26], the binary mask information in memory frames is embedded into $V$ with an additional memory encoder network and thus can also be propagated to the current frame by using Eq. 2. A convolutional decoder network following the propagated feature will decode the aggregated feature and predict the single-object probability logit of the current frame.

The main problem of propagating and decoding multi-object mask information in an end-to-end network is how to adapt the network to different target numbers. To overcome this problem, we propose an identification mechanism consisting of identification embedding and decoding based on attention mechanisms.

First, an **Identification Embedding** mechanism is proposed to embed the masks of multiple different targets into the same feature space for propagation. As seen in Fig. 2b, we initialize an identity bank, $D \in \mathbb{R}^{M \times C}$, where $M$ identification vectors with $C$ dimensions are stored. For embedding multiple different target masks, each target will be randomly assigned a different identification vector. Assuming $N$ ($N < M$) targets are in the video scenery, the formula of embedding the targets' one-hot mask, $Y \in \{0, 1\}^{THW \times N}$, into a identification embedding, $E \in \mathbb{R}^{THW \times C}$, by randomly assigning identification vector from the bank $D$ is,

$$E = ID(Y, D) = YPD, \tag{3}$$

where $P \in \{0, 1\}^{N \times M}$ is a random permutation matrix, satisfying that $P^{tr}P$ is equal to a $M \times M$ unit matrix, for randomly selecting $N$ identification embeddings. After the $ID$ assignment, different target has different identification embedding, and thus we can propagate all the target identification information from memory frames to the current frame by attaching the identification embedding $E$ with the attention value $V$, *i.e.*,

$$V' = AttID(Q, K, V, Y|D) = Att(Q, K, V + ID(Y, D)) = Att(Q, K, V + E), \tag{4}$$

where $V' \in \mathbb{R}^{HW \times C}$ aggregates all the multiple targets' embeddings from the propagation.

For **Identification Decoding**, *i.e.*, predicting all the targets' probabilities from the aggregated feature $V'$, we firstly predict the probability logit for every identity in the bank $D$ by employing a convolutional decoding network $F^{\mathcal{D}}$, and then select the assigned ones and calculate the probabilities, *i.e.*,

$$Y' = softmax(PF^{\mathcal{D}}(V')) = softmax(PL^D), \tag{5}$$

where $L^D \in \mathbb{R}^{HW \times M}$ is all the $M$ identities' probability logits, $P$ is the same as the selecting matrix used in the identity assignment (Eq. 3), and $Y' \in [0, 1]^{HW \times N}$ is the probability prediction of all the $N$ targets.

For training, common multi-class segmentation losses, such as cross-entropy loss, can be used to optimize the multi-object $Y'$ regarding the ground-truth labels. The identity bank $D$ is trainable and randomly initialized at the training beginning. To ensure that all the identification vectors have the same opportunity to compete with each other, we randomly reinitialize the identification selecting matrix $P$ in each video sample and each optimization iteration.

## 4.2 Long Short-Term Transformer for Hierarchical Matching and Propagation

Previous methods [26, 34] always utilize only one layer of attention (Eq. 2) to aggregate single-object information. In our identification-based multi-object pipeline, we found that a single attention layer cannot fully model multi-object association, which naturally should be more complicated than single-object processes. Thus, we consider constructing hierarchical matching and propagation by using a series of attention layers. Recently, transformer blocks [39] have been demonstrated to be stable and promising in constructing hierarchical attention structures in visual tasks [8, 12]. Based on transformer blocks, we carefully design a Long Short-Term Transformer (LSTT) block for multi-object VOS.

Following the common transformer blocks [39, 11], LSTT firstly employs a self-attention layer, which is responsible for learning the association or correlation among the targets within the current frame. Then, LSTT additionally introduces a long-term attention, for aggregating targets' information from long-term memory frames and a short-term attention, for learning temporal smoothness from nearby short-term frames. The final module is based on a common 2-layer feed-forward MLP with

GELU [17] non-linearity in between. Fig. 2c shows the structure of an LSTT block. Notably, all these attention modules are implemented in the form of the multi-head attention [39], *i.e.*, multiple attention modules followed by concatenation and a linear projection. Nevertheless, we only introduce their single-head formulas below for the sake of simplicity.

**Long-Term Attention** is responsible for aggregating targets' information from past memory frames, which contains the reference frame and stored predicted frames, to the current frame. Since the time intervals between the current frame and past frames are variable and can be long-term, the temporal smoothness is difficult to guarantee. Thus, the long-term attention employs non-local attention like Eq. 2. Let $X_l^t \in \mathbb{R}^{HW \times C}$ denotes the input feature embedding at time $t$ and in block $l$, where $l \in \{1, ..., L\}$ is the block index of LSTT, the formula of the long-term attention is,

$$AttLT(X_l^t, X_l^{\mathbf{m}}, Y^{\mathbf{m}}) = AttID(X_l^t W_l^K, X_l^{\mathbf{m}} W_l^K, X_l^{\mathbf{m}} W_l^V, Y^{\mathbf{m}} | D), \qquad (6)$$

where $X_l^{\mathbf{m}} = Concat(X_l^{m_1}, ..., X_l^{m_T})$ and $Y^{\mathbf{m}} = Concat(Y^{m_1}, ..., Y^{m_T})$ are the input feature embeddings and target masks of memory frames with indices $\mathbf{m} = \{m_1, ..., m_T\}$. Besides, $W_l^K \in \mathbb{R}^{C \times C_k}$ and $W_l^V \in \mathbb{R}^{C \times C_v}$ are trainable parameters of the space projections for matching and propagation, respectively. Instead of using different projections for $X_l^t$ and $X_l^{\mathbf{m}}$, we found the training of LSTT is more stable with a siamese-like matching, *i.e.*, matching between the features within the same embedding space ($l$-th features with the same projection of $W_l^K$).

**Short-Term Attention** is employed for aggregating information in a spatial-temporal neighbourhood for each current-frame location. Intuitively, the image changes across several contiguous video frames are always smooth and continuous. Thus, the target matching and propagation in contiguous frames can be restricted in a small spatial-temporal neighborhood, leading to better efficiency than non-local processes. Considering $n$ neighbouring frames with indices $\mathbf{n} = \{t - 1, ..., t - n\}$ are in the spatial-temporal neighbourhood, the features and masks of these frames are $X_l^{\mathbf{n}} = Concat(X_l^{t-1}, ..., X_l^{t-n})$ and $Y^{\mathbf{n}} = Concat(Y^{t-1}, ..., Y^{t-n})$, and then the formula of the short-term attention at each spatial location $p$ is,

$$AttST(X_l^t, X_l^{\mathbf{n}}, Y^{\mathbf{n}} | p) = AttLT(X_{l,p}^t, X_{l,\mathcal{N}(p)}^{\mathbf{n}}, Y_{l,\mathcal{N}(p)}^{\mathbf{n}}), \qquad (7)$$

where $X_{l,p}^t \in \mathbb{R}^{1 \times C}$ is the feature of $X_l^t$ at location $p$, $\mathcal{N}(p)$ is a $\lambda \times \lambda$ spatial neighbourhood centered at location $p$, and thus $X_{l,\mathcal{N}(p)}^{\mathbf{n}}$ and $Y_{l,\mathcal{N}(p)}^{\mathbf{n}}$ are the features and masks of the spatial-temporal neighbourhood, respectively, with a shape of $n\lambda^2 \times C$ or $n\lambda^2 \times N$.

When extracting features of the first frame $t = 1$, there is no memory frames or previous frames, and hence we use $X_l^1$ to replace $X_l^{\mathbf{m}}$ and $X_l^{\mathbf{n}}$. In other words, the long-term attention and the short-term attention are changed into self-attentions without adjusting the network structures and parameters.

## 5 Implementation Details

**Network Details:** For sufficiently validating the effectiveness of our identification mechanism and LSTT, we mainly use light-weight backbone encoder, MobileNet-V2 [33], and decoder, FPN [20] with Group Normalization [45]. The spatial neighborhood size $\lambda$ is set to 15, and the number of identification vectors, $M$, is set to 10, which is consistent with the maximum object number in the benchmarks [48, 31]. AOT performs well with PaddlePaddle [1] and PyTorch [28]. More details can be found in the supplementary material.

**Architecture Variants:** We build several AOT variant networks with different LSTT layer number $L$ or long-term memory size $\mathbf{m}$. The hyper-parameters of these variants are: **(1) AOT-Tiny**: $L = 1$, $\mathbf{m} = \{1\}$; **(2) AOT-Small**: $L = 2$, $\mathbf{m} = \{1\}$; **(3) AOT-Base**: $L = 3$, $\mathbf{m} = \{1\}$; **(4) AOT-Large**: $L = 3$, $\mathbf{m} = \{1, 1 + \delta, 1 + 2\delta, 1 + 3\delta, ...\}$. In the experiments, we also equip AOT-L with ResNet50 (R50) [16] or Swin-B [22].

AOT-S is a small model with only 2 layers of LSTT block. Compared to AOT-S, AOT-T utilizes only 1 layer of LSTT, and AOT-B/L uses 3 layers. In AOT-T/S/B, only the first frame is considered into long-term memory, which is similar to [41, 50], leading to a smooth efficiency. In AOT-L, the predicted frames are stored into long-term memory per $\delta$ frames, following the memory reading strategy [26]. We set $\delta$ to 2/5 for training/testing.

**Training Details:** Following [46, 26, 23, 34], the training stage is divided into two phases: (1) pre-training on sythetic video sequence generated from static image datasets [14, 21, 10, 36, 15] by

Table 1: The quantitative evaluation on multi-object benchmarks, YouTube-VOS [48] and DAVIS 2017 [31]. **Y**: using YouTube-VOS for training. $^{*}$: using 600p instead of 480p videos in inference. $^{\ddagger}$: timing extrapolated from single-object speed assuming linear scaling in the number of objects.

(a) YouTube-VOS

| | | Seen | | Unseen | | |
|---|---|---|---|---|---|---|
| Methods | $\mathcal{J}\&\mathcal{F}$ | $\mathcal{J}$ | $\mathcal{F}$ | $\mathcal{J}$ | $\mathcal{F}$ | FPS |
| *Validation 2018 Split* | | | | | | |
| STM[ICCV19] [26] | 79.4 | 79.7 | 84.2 | 72.8 | 80.9 | - |
| KMN[ECCV20] [34] | 81.4 | 81.4 | 85.6 | 75.3 | 83.3 | - |
| CFBI[ECCV20] [50] | 81.4 | 81.1 | 85.8 | 75.3 | 83.4 | 3.4 |
| LWL[ECCV20] [6] | 81.5 | 80.4 | 84.9 | 76.4 | 84.4 | - |
| SST[CVPR21] [13] | 81.7 | 81.2 | - | 76.0 | - | - |
| CFBI+[TPAMI21] [51] | 82.8 | 81.8 | 86.6 | 77.1 | 85.6 | 4.0 |
| AOT-T | 80.2 | 80.1 | 84.5 | 74.0 | 82.2 | **41.0** |
| AOT-S | 82.6 | 82.0 | 86.7 | 76.6 | 85.0 | 27.1 |
| AOT-B | 83.5 | 82.6 | 87.5 | 77.7 | 86.0 | 20.5 |
| AOT-L | 83.8 | 82.9 | 87.9 | 77.7 | **86.5** | 16.0 |
| R50-AOT-L | 84.1 | 83.7 | 88.5 | **78.1** | 86.1 | 14.9 |
| SwinB-AOT-L | **84.5** | **84.3** | **89.3** | 77.9 | 86.4 | 9.3 |
| *Validation 2019 Split* | | | | | | |
| CFBI[ECCV20] [50] | 81.0 | 80.6 | 85.1 | 75.2 | 83.0 | 3.4 |
| SST[CVPR21] [13] | 81.8 | 80.9 | - | 76.6 | - | - |
| CFBI+[TPAMI21] [51] | 82.6 | 81.7 | 86.2 | 77.1 | 85.2 | 4.0 |
| AOT-T | 79.7 | 79.6 | 83.8 | 73.7 | 81.8 | **41.0** |
| AOT-S | 82.2 | 81.3 | 85.9 | 76.6 | 84.9 | 27.1 |
| AOT-B | 83.3 | 82.4 | 87.1 | 77.8 | 86.0 | 20.5 |
| AOT-L | 83.7 | 82.8 | 87.5 | 78.0 | **86.7** | 16.0 |
| R50-AOT-L | 84.1 | 83.5 | 88.1 | **78.4** | 86.3 | 14.9 |
| SwinB-AOT-L | **84.5** | **84.0** | **88.8** | **78.4** | **86.7** | 9.3 |

(b) DAVIS 2017

| Methods | $\mathcal{J}\&\mathcal{F}$ | $\mathcal{J}$ | $\mathcal{F}$ | FPS |
|---|---|---|---|---|
| *Validation 2017 Split* | | | | |
| CFBI [50] (**Y**) | 81.9 | 79.3 | 84.5 | 5.9 |
| SST [13] (**Y**) | 82.5 | 79.9 | 85.1 | - |
| KMN [34] | 76.0 | 74.2 | 77.8 | 4.2$^{\ddagger}$ |
| KMN [34] (**Y**) | 82.8 | 80.0 | 85.6 | 4.2$^{\ddagger}$ |
| CFBI+ [51] (**Y**) | 82.9 | 80.1 | 85.7 | 5.6 |
| AOT-T (**Y**) | 79.9 | 77.4 | 82.3 | **51.4** |
| AOT-S | 79.2 | 76.4 | 82.0 | 40.0 |
| AOT-S (**Y**) | 81.3 | 78.7 | 83.9 | 40.0 |
| AOT-B (**Y**) | 82.5 | 79.7 | 85.2 | 29.6 |
| AOT-L (**Y**) | 83.8 | 81.1 | 86.4 | 18.7 |
| R50-AOT-L (**Y**) | 84.9 | 82.3 | 87.5 | 18.0 |
| SwinB-AOT-L (**Y**) | **85.4** | **82.4** | **88.4** | 12.1 |
| *Testing 2017 Split* | | | | |
| CFBI [50] (**Y**) | 75.0 | 71.4 | 78.7 | 5.3 |
| CFBI$^{*}$ [50] (**Y**) | 76.6 | 73.0 | 80.1 | 2.9 |
| KMN$^{*}$ [34] (**Y**) | 77.2 | 74.1 | 80.3 | - |
| CFBI+$^{*}$ [51] (**Y**) | 78.0 | 74.4 | 81.6 | 3.4 |
| AOT-T (**Y**) | 72.0 | 68.3 | 75.7 | **51.4** |
| AOT-S (**Y**) | 73.9 | 70.3 | 77.5 | 40.0 |
| AOT-B (**Y**) | 75.5 | 71.6 | 79.3 | 29.6 |
| AOT-L (**Y**) | 78.3 | 74.3 | 82.3 | 18.7 |
| R50-AOT-L (**Y**) | 79.6 | 75.9 | 83.3 | 18.0 |
| SwinB-AOT-L (**Y**) | **81.2** | **77.3** | **85.1** | 12.1 |

randomly applying multiple image augmentations [46]. (2) main training on the VOS benchmarks [48, 31] by randomly applying video augmentations [50]. More details are in the supplementary material.

## 6 Experimental Results

We evaluate AOT on popular multi-object benchmarks, YouTube-VOS [48] and DAVIS 2017 [31], and single-object benchmark, DAVIS 2016 [30]. For YouTube-VOS experiments, we train our models on the YouTube-VOS 2019 training split. For DAVIS, we train on the DAVIS-2017 training split. When evaluating YouTube-VOS, we use the default 6FPS videos, and all the videos are restricted to be smaller than $1.3 \times 480p$ resolution. As to DAVIS, the default 480p 24FPS videos are used.

The evaluation metric is the $\mathcal{J}$ score, calculated as the average Intersect over Union (IoU) score between the prediction and the ground truth mask, and the $\mathcal{F}$ score, calculated as an average boundary similarity measure between the boundary of the prediction and the ground truth, and their mean value, denoted as $\mathcal{J}\&\mathcal{F}$. We evaluate all the results on official evaluation servers or with official tools.

### 6.1 Compare with the State-of-the-art Methods

**YouTube-VOS** [48] is the latest large-scale benchmark for multi-object video segmentation and is about 37 times larger than DAVIS 2017 (120 videos). Specifically, YouTube-VOS contains 3471 videos in the training split with 65 categories and 474/507 videos in the validation 2018/2019 split with additional 26 unseen categories. The unseen categories do not exist in the training split to evaluate algorithms' generalization ability.

As shown in Table 1a, AOT variants achieve superior performance on YouTube-VOS compared to the previous state-of-the-art methods. With our identification mechanism, AOT-S (82.6% $\mathcal{J}\&\mathcal{F}$)

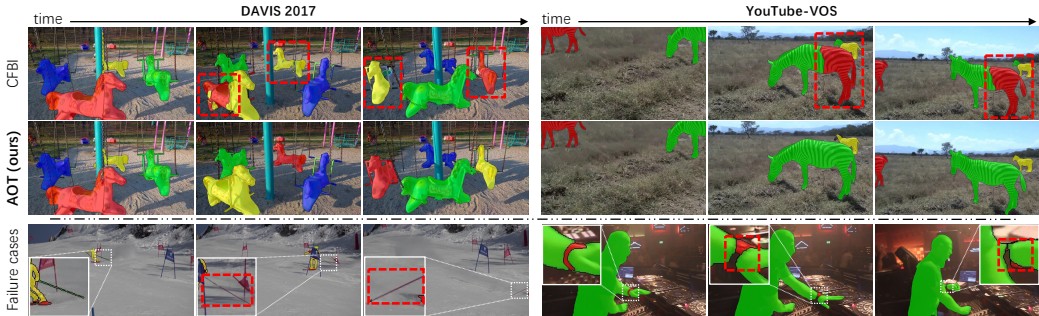

Figure 3: Qualitative results. (top) Compared with CFBI [50], AOT performs better when segmenting multiple highly similar objects (*carousels* and *zebras*). (bottom) AOT fails to segment some tiny objects (*ski poles* and *watch*) since AOT has no specific design for processing rare tiny objects.

is comparable with CFBI+ [51] (82.8%) while running about 7× faster (27.1 *vs* 4.0FPS). By using more LSTT blocks, AOT-B improves the performance to 83.5%. Moreover, AOT-L further improves both the seen and unseen scores by utilizing the memory reading strategy, and our R50-AOT-L (**84.1%/84.1%**) significantly outperforms the previous methods on the validation 2018/2019 split while maintaining an efficient speed (14.9FPS).

**DAVIS 2017** [31] is a multi-object extension of DAVIS 2016. The validation split of DAVIS 2017 consists of 30 videos with 59 objects, and the training split contains 60 videos with 138 objects. Moreover, the testing split contains 30 more challenging videos with 89 objects in total.

Table 1b shows that our R50-AOT-L (**Y**) surpasses all the competitors on both the DAVIS-2017 validation (**84.9%**) and testing (**79.6%**) splits and maintains an efficient speed (18.0FPS). Notably, such a multi-object speed is the same as our single-object speed on DAVIS 2016. For the first time, processing mul-tiple objects can be as efficient as processing a single

Table 2: The quantitative evaluation on the single-object DAVIS 2016 [30].

| Methods | $\mathcal{J}\&\mathcal{F}$ | $\mathcal{J}$ | $\mathcal{F}$ | FPS |
|---|---|---|---|---|
| CFBI+ [51] (**Y**) | 89.9 | 88.7 | 91.1 | 5.9 |
| KMN [34] (**Y**) | 90.5 | 89.5 | 91.5 | 8.3 |
| AOT-T (**Y**) | 86.8 | 86.1 | 87.4 | **51.4** |
| AOT-S (**Y**) | 89.4 | 88.6 | 90.2 | 40.0 |
| AOT-B (**Y**) | 89.9 | 88.7 | 91.1 | 29.6 |
| AOT-L (**Y**) | 90.4 | 89.6 | 91.1 | 18.7 |
| R50-AOT-L (**Y**) | 91.1 | 90.1 | 92.1 | 18.0 |
| SwinB-AOT-L (**Y**) | **92.0** | **90.7** | **93.3** | 12.1 |

object over the AOT framework. We also evaluate our method without training with YouTube-VOS, and AOT-S (79.2%) performs much better than KMN [34] (76.0%) by +3.2%.

**DAVIS 2016** [30] is a single-object benchmark containing 20 videos in the validation split. Although our AOT aims at improving multi-object video segmentation, we also achieve a new state-of-the-art performance on DAVIS 2016 (R50-AOT-L (**Y**), **91.1%**). Under single-object scenarios, the multi-object superiority of AOT is limited, but R50-AOT-L still maintains an about 2× efficiency compared to KMN (18.0 *vs* 8.3FPS). Furthermore, our smaller variant, AOT-B (89.9%), achieves comparable performance with CFBI+ (89.9%) while running 5× faster (29.6 *vs* 5.9FPS).

Apart from the above results, replacing the AOT encoder from commonly used ResNet50 to SwinB can further boost our performance to higher level (Table 1a, 1b, and 2).

**Qualitative results:** Fig. 3 visualizes some qualitative results in comparison with CFBI [50], which only associates each object with its relative background. As demonstrated, CFBI is easier to confuse multiple highly similar objects. In contrast, our AOT tracks and segments all the targets accurately by associating all the objects uniformly. However, AOT fails to segment some tiny objects (*ski poles* and *watch*) since we do not make special designs for tiny objects.

## 6.2 Ablation Study

In this section, we analyze the main components and hyper-parameters of AOT and evaluate their impact on the VOS performance in Table 3.

**Identity number:** The number of the identification vectors, $M$, have to be larger than the object number in videos. Thus, we set $M$ to 10 in default to be consistent with the maximum object number

Table 3: Ablation study. The experiments are based on AOT-S and conducted on the validation 2018 split of YouTube-VOS [48] without pre-training on synthetic videos. Self: the position embedding type used in the self-attention. Rel: use relative positional embedding [35] on the local attention.

(a) Identity number

| $M$ | $\mathcal{J}\&\mathcal{F}$ | $\mathcal{J}^{seen}$ | $\mathcal{J}^{unseen}$ |
|---|---|---|---|
| 10 | **80.3** | **80.6** | **73.7** |
| 15 | 79.0 | 79.4 | 72.1 |
| 20 | 78.3 | 79.4 | 70.8 |
| 30 | 77.2 | 78.5 | 70.2 |

(b) Local window size

| $\lambda$ | $\mathcal{J}\&\mathcal{F}$ | $\mathcal{J}^{seen}$ | $\mathcal{J}^{unseen}$ |
|---|---|---|---|
| 15 | **80.3** | **80.6** | **73.7** |
| 11 | 78.8 | 79.5 | 71.9 |
| 7 | 78.3 | 79.3 | 70.9 |
| 0 | 74.3 | 74.9 | 67.6 |

(c) Local frame number

| $n$ | $\mathcal{J}\&\mathcal{F}$ | $\mathcal{J}^{seen}$ | $\mathcal{J}^{unseen}$ |
|---|---|---|---|
| 1 | **80.3** | **80.6** | **73.7** |
| 2 | 80.0 | 79.8 | 73.7 |
| 3 | 79.1 | 80.0 | 72.2 |
| 0 | 74.3 | 74.9 | 67.6 |

(d) LSTT block number

| $L$ | $\mathcal{J}\&\mathcal{F}$ | $\mathcal{J}^{seen}$ | $\mathcal{J}^{unseen}$ | FPS | Param |
|---|---|---|---|---|---|
| 2 | 80.3 | 80.6 | 73.7 | 27.1 | 7.0M |
| 3 | **80.9** | **81.1** | **74.0** | 20.5 | 8.3M |
| 1 | 77.9 | 78.8 | 71.0 | **41.0** | **5.7M** |

(e) Positional embedding

| Self | Rel | $\mathcal{J}\&\mathcal{F}$ | $\mathcal{J}^{seen}$ | $\mathcal{J}^{unseen}$ |
|---|---|---|---|---|
| sine | ✓ | **80.3** | **80.6** | **73.7** |
| none | ✓ | 80.1 | 80.4 | 73.5 |
| sine | - | 79.7 | 80.1 | 72.9 |

in the benchmarks [48, 31]. As seen in Table 3a, $M$ larger than 10 leads to worse performance since (1) no training video contains so many objects; (2) embedding more than 10 objects into the space with only 256 dimensions is difficult.

**Local window size:** Table 3b shows that larger local window size, $\lambda$, results in better performance. Without the local attention, $\lambda = 0$, the performance of AOT significantly drops from 80.3% to 74.3%, which demonstrates the necessity of the local attention.

**Local frame number:** In Table 3c, we also try to employ more previous frames in the local attention, but using only the $t-1$ frame (80.3%) performs better than using 2/3 frames (80.0%/79.1%). A possible reason is that the longer the temporal interval, the more intense the motion between frames, so it is easier to introduce more errors in the local matching when using an earlier previous frame.

**LSTT block number:** As shown in Table 3d, the AOT performance increases by using more LSTT blocks. Notably, the AOT with only one LSTT block (77.9%) reaches a fast real-time speed (41.0FPS) on YouTube-VOS, although the performance is -2.4% worse than AOT-S (80.3%). By adjusting the LSTT block number, we can flexibly balance the accuracy and speed of AOT.

**Position embedding:** In our default setting, we apply fixed sine spatial positional embedding to the self-attention following [8], and our local attention is equipped with learned relative positional embedding [35]. The ablation study is shown in Table 3e, where removing the sine embedding decreases the performance to 80.1% slightly. In contrast, the relative embedding is more important than the sine embedding. Without the relative embedding, the performance drops to 79.7%, which means the motion relationship between adjacent frames is helpful for local attention. We also tried to apply learned positional embedding to self-attention modules, but no positive effect was observed.

## 7 Conclusion

This paper proposes a novel and efficient approach for video object segmentation by associating objects with transformers and achieves superior performance on three popular benchmarks. A simple yet effective identification mechanism is proposed to associate, match, and decode all the objects uniformly under multi-object scenarios. For the first time, processing multiple objects in VOS can be efficient as processing a single object by using the identification mechanism. In addition, a long short-term transformer is designed for constructing hierarchical object matching and propagation for VOS. The hierarchical structure allows us to flexibly balance AOT between real-time speed and state-of-the-art performance by adjusting the LSTT number. We hope the identification mechanism will help ease the future study of multi-object VOS and related tasks (*e.g.*, video instance segmentation, interactive VOS, and multi-object tracking), and AOT will serve as a solid baseline.

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
