# Associating Objects with Transformers for Video Object Segmentation

**Zongxin Yang**[1,2], **Yunchao Wei**[3,4], **Yi Yang**[1]

[1] CCAI, College of Computer Science and Technology, Zhejiang University  [2] Baidu Research
[3] Institute of Information Science, Beijing Jiaotong University
[4] Beijing Key Laboratory of Advanced Information Science and Network
{zongxinyang1996, wychao1987, yee.i.yang}@gmail.com

## A   Appendix

### A.1   More Implementation Details

#### A.1.1   Network Details

For MobileNet-V2 encoder, we increase the final resolution of the encoder to $1/16$ by adding a dilation to the last stage and removing a stride from the first convolution of this stage. For ResNet-50 and SwinB encoders, we remove the last stage directly. The encoder features are flattened into sequences before LSTT. In LSTT, the input channel dimension is 256, and the head number is set to 8 for all the attention modules. To increase the receptive field of LSTT, we insert a depth-wise convolution layer with a kernel size of 5 between two layers of each feed-forward module. In our default setting of the short-term memory $\mathbf{n}$, only the previous $(t-1)$ frame is considered, which is similar to the local matching strategy [23, 33]. After LSTT, all the output features of LSTT blocks are reshaped into 2D shapes and will serve as the decoder input. Then, the FPN decoder progressively increases the feature resolution from $1/16$ to $1/4$ and decreases the channel dimension from 256 to 128 before the final output layer, which is used for identification decoding.

**Patch-wise Identity Bank:** Since the spatial size of LSTT features is only 1/16 of the input video, we can not directly assign identities to the pixels of high-resolution input mask to construct a low-resolution identification embedding. To overcome this problem, we further propose a strategy named patch-wise identity bank. In detail, we first separate the input mask into non-overlapping patches of $16 \times 16$ pixels. The original identity bank with $M$ identities is also expanded to a patch-wise identity bank, in which each identity has $16 \times 16$ sub-identity vectors corresponding to $16 \times 16$ positions in a patch. Hence, the pixels of an object region with different patch positions will have different sub-identity vectors under an assigned identity. By summing all the assigned sub-identities in each patch, we can directly construct a low-resolution identification embedding while keeping the shape information inside each patch.

#### A.1.2   Training Details

All the videos are firstly down-sampled to 480p resolution, and the cropped window size is $465 \times 465$. For optimization, we adopt the AdamW [11] optimizer and the sequential training strategy [33], whose sequence length is set to 5. The loss function is a 0.5:0.5 combination of bootstrapped cross-entropy loss and soft Jaccard loss [16]. For stabilizing the training, the statistics of BN [8] modules and the first two stages in the encoder are frozen, and Exponential Moving Average (EMA) [20] is used. Besides, we apply stochastic depth [7] to the self-attention and the feed-forward modules in LSTT.

35th Conference on Neural Information Processing Systems (NeurIPS 2021).

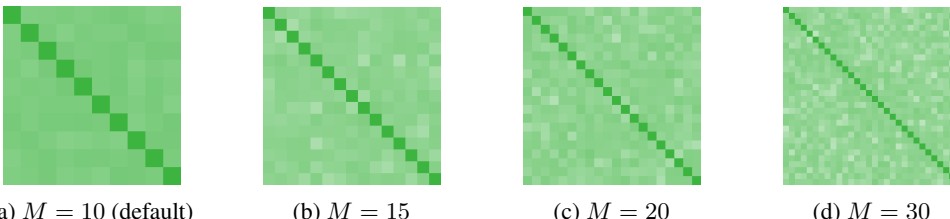

(a) $M = 10$ (default)     (b) $M = 15$     (c) $M = 20$     (d) $M = 30$

Figure 1: Visualization of the cosine similarity between every two of $M$ identification vectors in the identity bank. We use the form of a $M \times M$ symmetric matrix to visualize all the cosine similarities, and the values on the diagonal are all equal to 1. The darker the green color, the higher the similarity. In the case of $M = 10$, the similarities are stable and balanced. As the vector number $M$ increases, The visualized matrix becomes less and less smooth, which means the similarities become unstable.

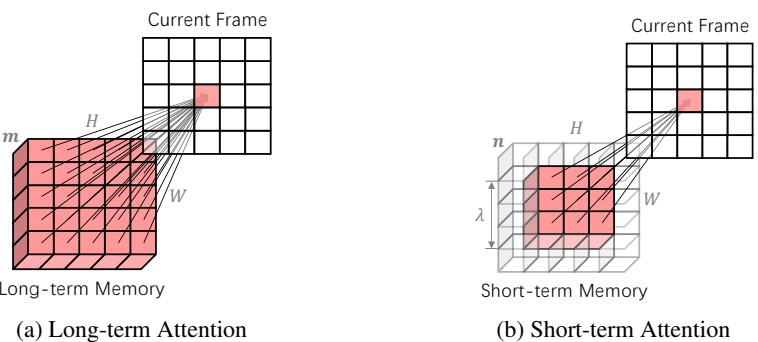

(a) Long-term Attention     (b) Short-term Attention

Figure 2: Illustrations of the long-term attention and the short-term attention. (a) The long-term attention employs a non-local manner to match all the locations in the long-term memory. (b) In contrast, the short-term attention only focus on a nearby spatial-temporal region with a shape of $n\lambda^2$.

The batch size is 16 and distributed on 4 Tesla V100 GPUs. For pre-training, we use an initial learning rate of $4 \times 10^{-4}$ and a weight decay of 0.03 for 100,000 steps. For main training, the initial learning rate is set to $2 \times 10^{-4}$ and the weight decay is 0.07. In addition, the training steps are 100,000 for YouTube-VOS or 50,000 for DAVIS. To relieve over-fitting, the initial learning rate of encoders is reduced to a 0.1 scale of other network parts. All the learning rates gradually decay to $2 \times 10^{-5}$ in a polynomial manner [33].

## A.2   Visualization of Identity Bank

In AOT, the identity bank is randomly initialized, and all the $M$ identification vectors are learned by being randomly assigned to objects during the training phase. Intuitively, all the identification vectors should be equidistant away from each other in the feature space because their roles are equivalent. To validate our hypothesis, we visualize the similarity between every two identification vectors in Fig. 1.

In our default setting, $M = 10$ (Fig. 1a), all the vectors are far away from each other, and the similarities remain almost the same. This phenomenon is consistent with our above hypothesis. In other words, the reliability and effectiveness of our identification mechanism are further verified.

In the ablation study, using more identities leads to worse results. To analyze the reason, we also visualize the learned identity banks with more vectors. Fig. 1b, 1c, and 1d demonstrate that maintaining equidistant between every two vectors becomes more difficult when the identity bank contains more vectors, especially when $M = 30$. There are two possible reasons for this phenomenon: (1) No training video contains enough objects to be assigned so many identities, and thus the network cannot learn to associate all the identities simultaneously; (2) the used space with only 256 dimensions is difficult for keeping more than 10 objects to be equidistant.

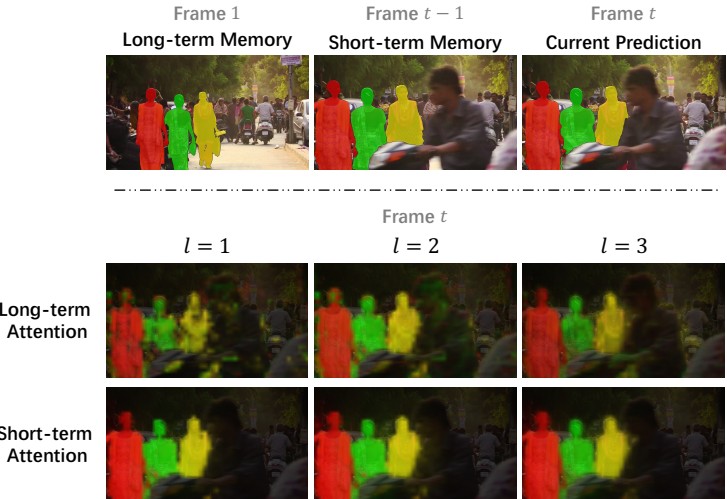

Figure 3: Visualization of long-term and short-term attention maps during the inference of DAVIS 2017 [21]. There are three similar people, marked in different colors, in the video. To sufficiently verify the effect of long-term attention, only the first frame is considered into the long-term memory, and thus we AOT-B to conduct the experiment. For visualization, we propagate the colored multi-object masks in long-term or short-term memory to the current frame regarding the corresponding attention map. The brighter the color, the stronger the attention. $l = 1$, $2$, and $3$ denote the 3 LSTT layers of AOT-B in order.

## A.3   Illustration of Long Short-term Attention

To facilitate understanding our long-term and short-term attention modules, we illustrate their processes in Fig. 2. Since the temporal smoothness between the current frame and long-term memory frames is difficult to guarantee, the long-term attention employs a non-local manner to match all the locations in the long-term memory. In contrast, short-term attention only focuses on a nearby spatial-temporal neighborhood of each current-frame location.

## A.4   Visualization of Hierarchical Matching and Propagation

In our AOT, we propose to construct a hierarchical framework, *i.e.*, LSTT, for multi-object matching and propagation, and the ablation study indicates that using more LSTT layers (or blocks) results in better VOS performance. To further validate the effectiveness of LSTT and analyze the behavior of each LSTT layer, we visualize long-term and short-term attention maps in each layer during inference, as shown in Fig. 3 and 4.

At the bottom of Fig. 3, the attention maps become more accurate and sharper as the index of layers increases. In the first layer, *i.e.*, $l = 1$, the current features have not aggregated the multi-object mask information from memory frames, and the long-term attention map is very vague and contains a lot of wrong matches among the objects and the background. Nevertheless, as the layer index increases, the mask information of all the objects is gradually aggregated so that the long-term attention becomes more and more accurate. Similarly, the quality, especially the boundary of objects, of the short-term attention improves as the layer index increases. Notably, the short-term attention performs well even in the first layer, $l = 1$, which is different from the long-term attention. The reason is that the neighborhood matching of short-term attention is easier than the long-term matching of long-term attention. However, long-term attention is still necessary because short-term attention will fail in some cases, such as occlusion (Fig. 4).

In short, the visual analysis further proves the necessity and effectiveness of our hierarchical LSTT. The hierarchical matching is not simply a combination of multiple matching processes. Critically, the multi-object information will be gradually aggregated and associated as the LSTT structure goes deeper, leading to more accurate attention-based matching.

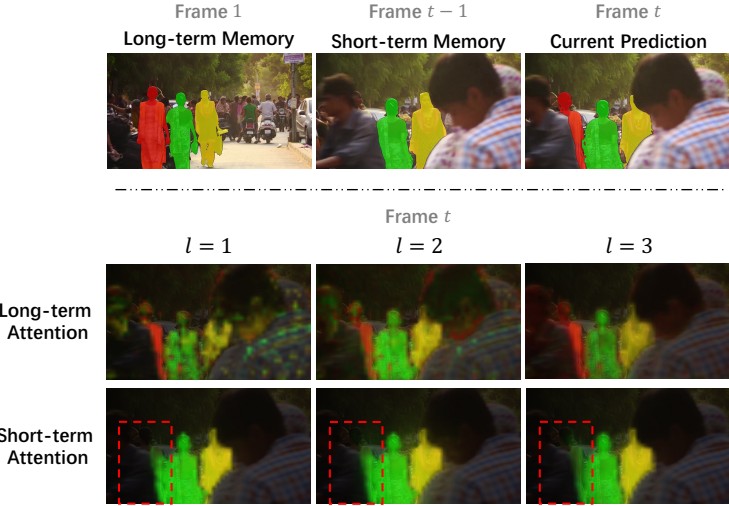

Figure 4: Visualization of long-term and short-term attention maps during the inference of DAVIS 2017 [21]. In this case, the red person is occluded in the $t-1$ frame, and thus the short-term attention fails to match the person in the current frame. However, the long-term attention generates an accurate attention map with a clean background in the last layer, $l = 3$, resulting in a correct prediction.

## A.5 Compare with More Methods

We compare our AOT with more VOS methods in Table 1 and 2. To compare with latest real-time methods [3, 10], we introduce the real-time AOT variant, *i.e.*, AOT-T.

Even on the single-object DAVIS 2016 [19], AOT-S (89.4%, 40.0FPS) can achieve comparable speed with SAT [3] (83.1%, 39FPS) and comparable performance with CFBI [33] (89.4%, 6.3FPS). On the multi-object DAVIS 2017 [21], AOT-T (79.9%, 51.4FPS) significantly outperforms SAT (72.3%, 19.5FPS) and GC (71.4%, 12.5FPS). Particularly, on the large-scale multi-object YouTube-VOS [29], AOT-T/S (80.2%/82.6%) achieves superior performance compared to previous real-time methods, SAT (63.6%) and GC (73.2%), while still maintaining a real-time speed (41.0FPS/27.1FPS).

## A.6 Additional Qualitative Results

We supply more qualitative results under multi-object scenarios on the large-scale YouTube-VOS [29] and the small-scale DAVIS 2017 [21] in Fig. 5 and 6, respectively. As demonstrated, our AOT-L is robust to many challenging VOS cases, including similar objects, occlusion, fast motion, and motion blur, etc.

## A.7 Border Impact and Future Works

The proposed AOT framework greatly simplifies the process of multi-object VOS and achieves a significant performance of effectiveness, robustness, and efficiency. Some AOT variants can achieve promising results while keeping real-time speed. In other words, AOT may promote the applications of VOS in real-time video systems, such as video conference, self-driving car, augmented reality, etc.

Nevertheless, the speed and accuracy of AOT can still be further improved. There is still a very large accuracy gap between the real-time AOT-T and the state-of-the-art SwinB-AOT-L. Moreover, AOT uses only a lightweight encoder and decoder. How to design stronger yet efficient encoders and decoders for VOS is still an open question.

As to related areas of VOS, the simple yet effective identification mechanism should also be promising for many tasks requiring multi-object matching, such as interactive video object segmentation [14, 17], video instance segmentation [1, 26, 31], and multi-object tracking [15, 27, 30]. Besides, the hierarchical LSTT may serve as a new solution for processing video representations in these tasks.

Table 1: Additional quantitative comparison on multi-object benchmarks, YouTube-VOS [29] and DAVIS 2017 [21].

(a) YouTube-VOS

| Methods | Seen | | | Unseen | | FPS |
|---|---|---|---|---|---|---|
| | $\mathcal{J}\&\mathcal{F}$ | $\mathcal{J}$ | $\mathcal{F}$ | $\mathcal{J}$ | $\mathcal{F}$ | |
| *Validation 2018 Split* | | | | | | |
| SAT[CVPR20] [3] | 63.6 | 67.1 | 70.2 | 55.3 | 61.7 | - |
| AG[CVPR19] [9] | 66.1 | 67.8 | - | 60.8 | - | - |
| PReM[ACCV18] [13] | 66.9 | 71.4 | 75.9 | 56.5 | 63.7 | 0.17 |
| BoLT[arXiv19] [25] | 71.1 | 71.6 | - | 64.3 | - | 0.74 |
| GC[ECCV20] [10] | 73.2 | 72.6 | 75.6 | 68.9 | 75.7 | - |
| STM[ICCV19] [18] | 79.4 | 79.7 | 84.2 | 72.8 | 80.9 | - |
| EGMN[ECCV20] [12] | 80.2 | 80.7 | 85.1 | 74.0 | 80.9 | - |
| KMN[ECCV20] [22] | 81.4 | 81.4 | 85.6 | 75.3 | 83.3 | - |
| CFBI[ECCV20] [33] | 81.4 | 81.1 | 85.8 | 75.3 | 83.4 | 3.4 |
| LWL[ECCV20] [2] | 81.5 | 80.4 | 84.9 | 76.4 | 84.4 | - |
| SST[CVPR21] [5] | 81.7 | 81.2 | - | 76.0 | - | - |
| CFBI+[TPAMI21] [34] | 82.8 | 81.8 | 86.6 | 77.1 | 85.6 | 4.0 |
| AOT-T | 80.2 | 80.1 | 84.5 | 74.0 | 82.2 | **41.0** |
| AOT-S | 82.6 | 82.0 | 86.7 | 76.6 | 85.0 | 27.1 |
| AOT-B | 83.5 | 82.6 | 87.5 | 77.7 | 86.0 | 20.5 |
| AOT-L | 83.8 | 82.9 | 87.9 | 77.7 | **86.5** | 16.0 |
| R50-AOT-L | 84.1 | 83.7 | 88.5 | **78.1** | 86.1 | 14.9 |
| SwinB-AOT-L | **84.5** | **84.3** | **89.3** | 77.9 | 86.4 | 9.3 |
| *Validation 2019 Split* | | | | | | |
| CFBI[ECCV20] [33] | 81.0 | 80.6 | 85.1 | 75.2 | 83.0 | 3.4 |
| SST[CVPR21] [5] | 81.8 | 80.9 | - | 76.6 | - | - |
| CFBI+[TPAMI21] [34] | 82.6 | 81.7 | 86.2 | 77.1 | 85.2 | 4.0 |
| AOT-T | 79.7 | 79.6 | 83.8 | 73.7 | 81.8 | **41.0** |
| AOT-S | 82.2 | 81.3 | 85.9 | 76.6 | 84.9 | 27.1 |
| AOT-B | 83.3 | 82.4 | 87.1 | 77.8 | 86.0 | 20.5 |
| AOT-L | 83.7 | 82.8 | 87.5 | 78.0 | **86.7** | 16.0 |
| R50-AOT-L | 84.1 | 83.5 | 88.1 | **78.4** | 86.3 | 14.9 |
| SwinB-AOT-L | **84.5** | **84.0** | **88.8** | **78.4** | **86.7** | 9.3 |

(b) DAVIS 2017

| Methods | $\mathcal{J}\&\mathcal{F}$ | $\mathcal{J}$ | $\mathcal{F}$ | FPS |
|---|---|---|---|---|
| *Validation 2017 Split* | | | | |
| OSMN [32] | 54.8 | 52.5 | 57.1 | 3.6‡ |
| VM [6] | 62.4 | 56.5 | 68.2 | 2.9 |
| OnA [24] | 63.6 | 61.0 | 66.1 | 0.04 |
| RGMP [28] | 66.7 | 64.8 | 68.6 | 3.6‡ |
| AG [9] (**Y**) | 70.0 | 67.2 | 72.7 | 7.1‡ |
| GC [10] | 71.4 | 69.3 | 73.5 | 12.5‡ |
| FEEL [23] (**Y**) | 71.5 | 69.1 | 74.0 | 2.0 |
| SAT [3] (**Y**) | 72.3 | 68.6 | 76.0 | 19.5‡ |
| PReM [13] | 77.8 | 73.9 | 81.7 | 0.03 |
| LWL [2] (**Y**) | 81.6 | 79.1 | 84.1 | 2.5‡ |
| STM [18] (**Y**) | 81.8 | 79.2 | 84.3 | 3.1‡ |
| CFBI [33] (**Y**) | 81.9 | 79.3 | 84.5 | 5.9 |
| SST [5] (**Y**) | 82.5 | 79.9 | 85.1 | - |
| EGMN [12] (**Y**) | 82.8 | 80.2 | 85.2 | 2.5‡ |
| KMN [22] | 76.0 | 74.2 | 77.8 | 4.2‡ |
| KMN [22] (**Y**) | 82.8 | 80.0 | 85.6 | 4.2‡ |
| CFBI+ [34] (**Y**) | 82.9 | 80.1 | 85.7 | 5.6 |
| AOT-T (**Y**) | 79.9 | 77.4 | 82.3 | **51.4** |
| AOT-S | 79.2 | 76.4 | 82.0 | 40.0 |
| AOT-S (**Y**) | 81.3 | 78.7 | 83.9 | 40.0 |
| AOT-B (**Y**) | 82.5 | 79.7 | 85.2 | 29.6 |
| AOT-L (**Y**) | 83.8 | 81.1 | 86.4 | 18.7 |
| R50-AOT-L (**Y**) | 84.9 | 82.3 | 87.5 | 18.0 |
| SwinB-AOT-L (**Y**) | **85.4** | **82.4** | **88.4** | 12.1 |
| *Testing 2017 Split* | | | | |
| OSMN [32] | 41.3 | 37.7 | 44.9 | 2.4‡ |
| RGMP [28] | 52.9 | 51.3 | 54.4 | 2.4‡ |
| OnA [24] | 56.5 | 53.4 | 59.6 | 0.03 |
| FEEL [23] (**Y**) | 57.8 | 55.2 | 60.5 | 1.9 |
| PReM [13] | 71.6 | 67.5 | 75.7 | 0.02 |
| STM* [18] (**Y**) | 72.2 | 69.3 | 75.2 | - |
| CFBI [33] (**Y**) | 75.0 | 71.4 | 78.7 | 5.3 |
| CFBI* [33] (**Y**) | 76.6 | 73.0 | 80.1 | 2.9 |
| KMN* [22] (**Y**) | 77.2 | 74.1 | 80.3 | - |
| CFBI+* [34] (**Y**) | 78.0 | 74.4 | 81.6 | 3.4 |
| AOT-T (**Y**) | 72.0 | 68.3 | 75.7 | **51.4** |
| AOT-S (**Y**) | 73.9 | 70.3 | 77.5 | 40.0 |
| AOT-B (**Y**) | 75.5 | 71.6 | 79.3 | 29.6 |
| AOT-L (**Y**) | 78.3 | 74.3 | 82.3 | 18.7 |
| R50-AOT-L (**Y**) | 79.6 | 75.9 | 83.3 | 18.0 |
| SwinB-AOT-L (**Y**) | **81.2** | **77.3** | **85.1** | 12.1 |

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

Table 2: Additional quantitative comparison on DAVIS 2016 [19].

| Methods | $\mathcal{J}\&\mathcal{F}$ | $\mathcal{J}$ | $\mathcal{F}$ | FPS |
|---|---|---|---|---|
| OSMN [32] | - | 74.0 | | 7.1 |
| PML [4] | 77.4 | 75.5 | 79.3 | 3.6 |
| VM [6] | 80.9 | 81.0 | 80.8 | 3.1 |
| FEEL [23] (**Y**) | 81.7 | 81.1 | 82.2 | 2.2 |
| RGMP [28] | 81.8 | 81.5 | 82.0 | 7.1 |
| AG [9] (**Y**) | 82.1 | 82.2 | 82.0 | 14.3 |
| SAT [3] (**Y**) | 83.1 | 82.6 | 83.6 | 39 |
| OnA [24] | 85.0 | 85.7 | 84.2 | 0.08 |
| GC [10] | 86.6 | 87.6 | 85.7 | 25 |
| PReM [13] | 86.8 | 84.9 | 88.6 | 0.03 |
| STM [18] (**Y**) | 89.3 | 88.7 | 89.9 | 6.3 |
| CFBI [33] (**Y**) | 89.4 | 88.3 | 90.5 | 6.3 |
| CFBI+ [34] (**Y**) | 89.9 | 88.7 | 91.1 | 5.9 |
| KMN [22] (**Y**) | 90.5 | 89.5 | 91.5 | 8.3 |
| AOT-T (**Y**) | 86.8 | 86.1 | 87.4 | **51.4** |
| AOT-S (**Y**) | 89.4 | 88.6 | 90.2 | 40.0 |
| AOT-B (**Y**) | 89.9 | 88.7 | 91.1 | 29.6 |
| AOT-L (**Y**) | 90.4 | 89.6 | 91.1 | 18.7 |
| R50-AOT-L (**Y**) | 91.1 | 90.1 | 92.1 | 18.0 |
| SwinB-AOT-L (**Y**) | **92.0** | **90.7** | **93.3** | 12.1 |

[4] Chen, Y., Pont-Tuset, J., Montes, A., Van Gool, L.: Blazingly fast video object segmentation with pixel-wise metric learning. In: CVPR. pp. 1189–1198 (2018)

[5] Duke, B., Ahmed, A., Wolf, C., Aarabi, P., Taylor, G.W.: Sstvos: Sparse spatiotemporal transformers for video object segmentation. In: CVPR (2021)

[6] Hu, Y.T., Huang, J.B., Schwing, A.G.: Videomatch: Matching based video object segmentation. In: ECCV. pp. 54–70 (2018)

[7] Huang, G., Sun, Y., Liu, Z., Sedra, D., Weinberger, K.Q.: Deep networks with stochastic depth. In: ECCV. pp. 646–661. Springer (2016)

[8] Ioffe, S., Szegedy, C.: Batch normalization: Accelerating deep network training by reducing internal covariate shift. In: ICML (2015)

[9] Johnander, J., Danelljan, M., Brissman, E., Khan, F.S., Felsberg, M.: A generative appearance model for end-to-end video object segmentation. In: CVPR. pp. 8953–8962 (2019)

[10] Li, Y., Shen, Z., Shan, Y.: Fast video object segmentation using the global context module. In: ECCV. pp. 735–750. Springer (2020)

[11] Loshchilov, I., Hutter, F.: Decoupled weight decay regularization. In: ICLR (2019)

[12] Lu, X., Wang, W., Danelljan, M., Zhou, T., Shen, J., Van Gool, L.: Video object segmentation with episodic graph memory networks. In: ECCV (2020)

[13] Luiten, J., Voigtlaender, P., Leibe, B.: Premvos: Proposal-generation, refinement and merging for video object segmentation. In: ACCV. pp. 565–580 (2018)

[14] Miao, J., Wei, Y., Yang, Y.: Memory aggregation networks for efficient interactive video object segmentation. In: CVPR (2020)

[15] Milan, A., Leal-Taixé, L., Reid, I., Roth, S., Schindler, K.: Mot16: A benchmark for multi-object tracking. arXiv preprint arXiv:1603.00831 (2016)

[16] Nowozin, S.: Optimal decisions from probabilistic models: the intersection-over-union case. In: CVPR. pp. 548–555 (2014)

[17] Oh, S.W., Lee, J.Y., Xu, N., Kim, S.J.: Fast user-guided video object segmentation by interaction-and-propagation networks. In: CVPR. pp. 5247–5256 (2019)

[18] Oh, S.W., Lee, J.Y., Xu, N., Kim, S.J.: Video object segmentation using space-time memory networks. In: ICCV (2019)

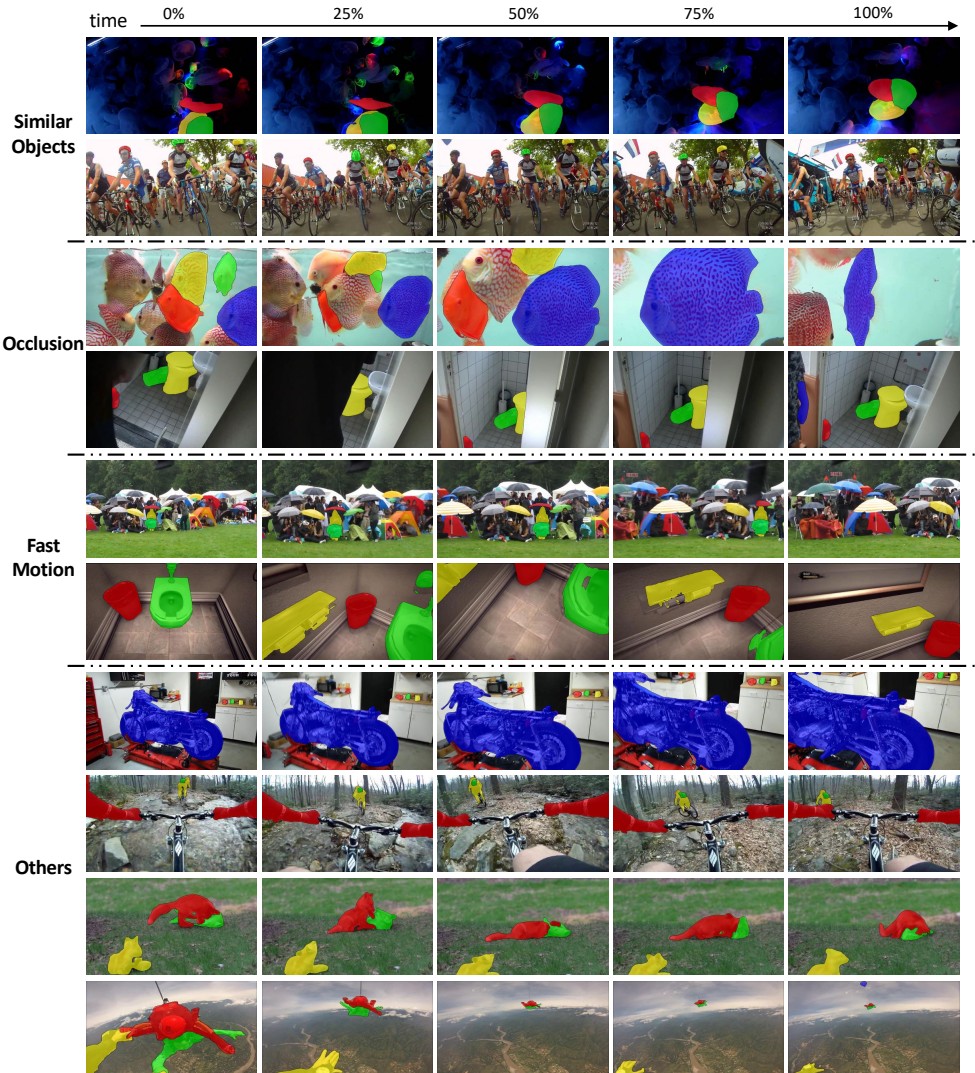

Figure 5: Qualitative results on the validation 2019 split of YouTube-VOS [29]. Our AOT-L performs well under many challenging multi-object cases, including similar objects, occlusion, and fast motion, etc.

[19] Perazzi, F., Pont-Tuset, J., McWilliams, B., Van Gool, L., Gross, M., Sorkine-Hornung, A.: A benchmark dataset and evaluation methodology for video object segmentation. In: CVPR. pp. 724–732 (2016)

[20] Polyak, B.T., Juditsky, A.B.: Acceleration of stochastic approximation by averaging. SIAM journal on control and optimization **30**(4), 838–855 (1992)

[21] Pont-Tuset, J., Perazzi, F., Caelles, S., Arbeláez, P., Sorkine-Hornung, A., Van Gool, L.: The 2017 davis challenge on video object segmentation. arXiv preprint arXiv:1704.00675 (2017)

[22] Seong, H., Hyun, J., Kim, E.: Kernelized memory network for video object segmentation. In: ECCV (2020)

[23] Voigtlaender, P., Chai, Y., Schroff, F., Adam, H., Leibe, B., Chen, L.C.: Feelvos: Fast end-to-end embedding learning for video object segmentation. In: CVPR. pp. 9481–9490 (2019)

[24] Voigtlaender, P., Leibe, B.: Online adaptation of convolutional neural networks for video object segmentation. In: BMVC (2017)

[25] Voigtlaender, P., Luiten, J., Leibe, B.: Boltvos: Box-level tracking for video object segmentation. arXiv preprint arXiv:1904.04552 (2019)

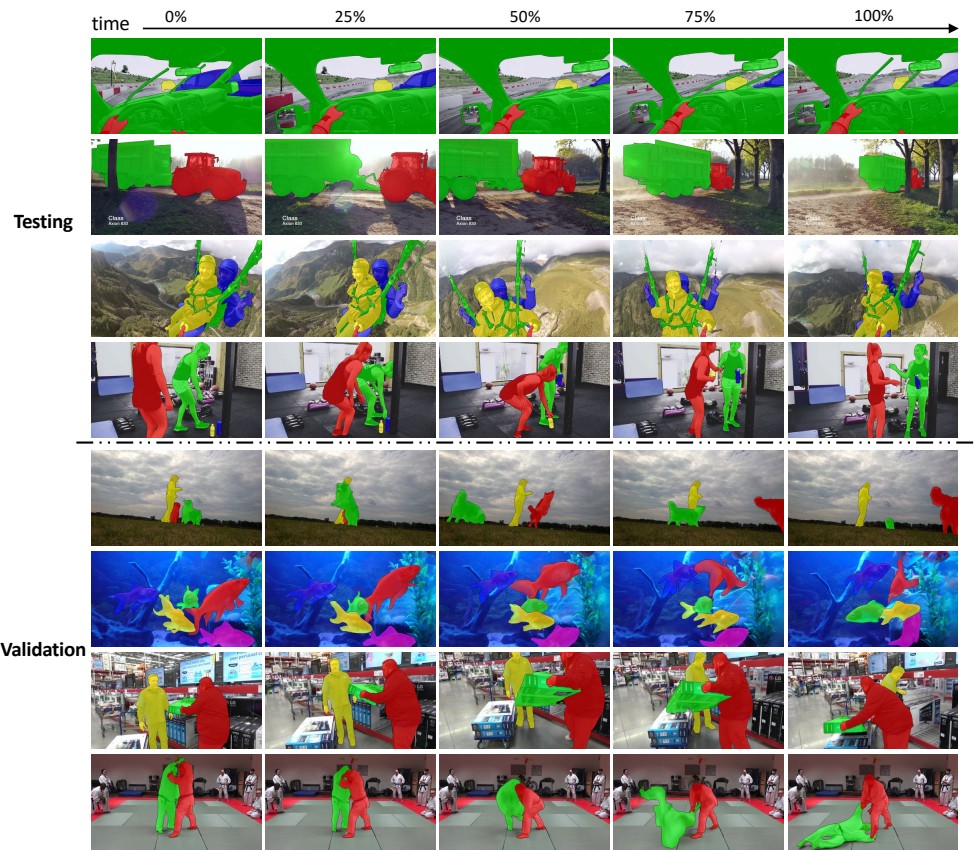

Figure 6: Qualitative results on DAVIS 2017 [21].

[26] Wang, Y., Xu, Z., Wang, X., Shen, C., Cheng, B., Shen, H., Xia, H.: End-to-end video instance segmentation with transformers. arXiv preprint arXiv:2011.14503 (2020)

[27] Wang, Z., Zheng, L., Liu, Y., Wang, S.: Towards real-time multi-object tracking. In: ECCV. Springer (2020)

[28] Wug Oh, S., Lee, J.Y., Sunkavalli, K., Joo Kim, S.: Fast video object segmentation by reference-guided mask propagation. In: CVPR. pp. 7376–7385 (2018)

[29] Xu, N., Yang, L., Fan, Y., Yue, D., Liang, Y., Yang, J., Huang, T.: Youtube-vos: A large-scale video object segmentation benchmark. arXiv preprint arXiv:1809.03327 (2018)

[30] Xu, Z., Zhang, W., Tan, X., Yang, W., Huang, H., Wen, S., Ding, E., Huang, L.: Segment as points for efficient online multi-object tracking and segmentation. In: ECCV. pp. 264–281. Springer (2020)

[31] Yang, L., Fan, Y., Xu, N.: Video instance segmentation. In: ICCV. pp. 5188–5197 (2019)

[32] Yang, L., Wang, Y., Xiong, X., Yang, J., Katsaggelos, A.K.: Efficient video object segmentation via network modulation. In: CVPR. pp. 6499–6507 (2018)

[33] Yang, Z., Wei, Y., Yang, Y.: Collaborative video object segmentation by foreground-background integration. In: ECCV (2020)

[34] Yang, Z., Wei, Y., Yang, Y.: Collaborative video object segmentation by multi-scale foreground-background integration. TPAMI (2021)