# OpenReview forum: "Associating Objects with Transformers for Video Object Segmentation"
_NeurIPS.cc/2021/Conference — NeurIPS 2021 Poster_

### Official Review · Reviewer_tUYE · 2021-07-15

**Rating:** 8
**Confidence:** 4

**Summary:**

The paper tackles the problem of video object segmentation. Firstly, the authors propose an identification mechanism which allows jointly segmenting multiple objects in a video with constant computational complexity w.r.t. the number of objects. This is achieved by mapping the masks of different objects to a common embedding space, which is then decoded by a single decoder. Secondly, the authors introduce a transformer based architecture with both long term (old frames) and short term (recent frames) matching to obtain accurate segmentation. The proposed approach obtains state-of-the-art results on both YouTubeVOS (83.7 $\mathcal{J}$&$\mathcal{F}$ on val2018) and DAVIS (83.0% $\mathcal{J}$&$\mathcal{F}$ on DAVIS 2017) benchmarks, while being computationally efficient.


**Limitations And Societal Impact:**

The authors discuss the failure cases of the proposed method.

**Main Review:**


### **Strengths**
S1) The proposed identification mechanism to handle multiple objects is novel and interesting. Compared to existing approaches, the proposed module allows segmenting multiple objects simultaneously without incurring additional computational costs compared to segmenting a single object.

S2) The second contribution of long short-term transformer based VOS architecture is also novel, interesting, and shown to be effective via appropriate ablation studies.

S3) The method obtains impressive results on standard VOS benchmarks YouTubeVOS and DAVIS, outperforming previous state-of-the-art by a significant margin.

S4) The paper is well-written and easy to read and understand.


### **Weaknesses**

W1) I do not think the use of term 'hierarchical matching' (used throughout the paper) is strictly correct. If I understand correctly, the authors stack multiple transformer blocks together to process the input frame, as commonly done in transformer based architectures. Referring to this as hierarchical matching seems confusing and slightly misleading. I would suggest omitting the use of 'hierarchical'.

W2) How are the FPS for other methods obtained in Table 1. Did the authors compute them using identical hardware as their method, or are they directly obtained from the corresponding papers? This should be clearly stated. Since the implementation of SOTA methods such as STM, CFBI, LWL are available, I would recommend running these methods using identical hardware in order to obtain a fairer speed comparison.


While not strictly necessary, I believe the paper will greatly benefit from additional experimental analysis of the proposed method, as listed below.

C1) The proposed approach obtains improved results compared to previous methods. However, it is not clear whether the improvement is mainly due to the long short-term transformer based architecture, or whether the joint segmentation of multiple targets enabled by the identification mechanism also leads to improved segmentation accuracy, in addition to computational efficiency. Thus, it would be interesting to evaluate a variant of the method which independently segments each object using the long short-term transformer, and then merges the masks using a soft-aggregation as done in STM.

C2) What is the impact of learning the identification vectors? What would be the performance drop if random identification vectors are used instead of the learned ones?

C3) The authors add the identification embedding of the target mask to the value computed using the image features. Did the authors experiment any other architectures to integrate the identity embedding into the value? For instance, is it possible to concatenate the identity embedding with the initial value and then pass it through a small network?

C4) What is the performance if only the short term (local) attention is used?


**Time Spent Reviewing:**

4

---

> ### Author Response · Authors · 2021-08-09
> **Rebuttal**
>
> We appreciate you taking the time to share your comments and suggestions in the review assessment. We are glad that you and all the other reviewers recognize our paper's novelty and contributions. All your comments are addressed point by point in the following.
>
> **Weaknesses**:
>
> **Q1**: The use of term "hierarchical matching."
>
> **A1**: The use of the term "hierarchical matching" is necessary to highlight two differences between our LSTT and previous methods.
>
> 1. Why use "matching"? Compared to common vision transformers based on self-attention, our LSTT focuses on another kind of attention, i.e., attention-based matching and propagation, which is responsible for propagating the visual features and mask features from past frames to the current frame. Unlike the symmetrical self-attention, our long-short term attention is asymmetric and has to be conditional on our identification embedding or other kinds of mask embeddings exclusive to VOS and related tasks.
>
> 2. Why use "hierarchical"? Previous attention-based VOS methods [3,4,5] use only one layer of the matching process. In contrast, our LSTT block or long-short term attention allows the construction of hierarchical matching structures effectively and efficiently. The main problem of constructing hierarchical matching is how to efficiently couple visual features and mask features during inference. When predicting one frame, previous methods first extract mask-agnostic visual features of this frame and then predict the current mask. For coupling the predicted mask with the visual features, they have to use another network to embedding the current frame with the predicted mask into mask-aware features again. This dual-network approach has been widely used by previous attention-based VOS methods but is not easy for constructing hierarchical matching. In contrast, we solve the dual-network problem by decoupling visual features and mask features with the help of our long-short term attention and identification mechanism.
>
> Compared to the single-layer matching of previous VOS methods [3,4,5], our "hierarchical matching" allows us to flexibly balance AOT between real-time speed and state-of-the-art performance by adjusting the matching layer number, as shown in Table 1. Furthermore, the visualization shown in the supplementary material (Fig. 3 and 4) demonstrates that our "hierarchical matching" can spontaneously learn a coarse-to-fine matching behavior, which is reasonable and promising. The structural excellence shown in the experimental results further shows the necessity to highlight the term "hierarchical matching."
>
> **Q2**: How are the FPS results for other methods obtained?
>
> **A2**: Following a common practice used by previous SOTA methods [3,4,5,6,7,25], we copied the FPS results from other methods' papers. The reason for this phenomenon is that very few VOS methods have released their codes. Within SOTA methods, CFBI's authors have released all the evaluation details. Thus, we recorded the FPS results of AOT variants under the same device environment as CFBI. And in our submission, we mainly compared our speed with CFBI for the sake of fairness.
>
> LWL's authors also released all the evaluation details. But we found they didn't follow the default setting of YouTube-VOS to evaluate their method. The default annotation rate of YouTube-VOS is 6FPS, but the reported performance of LWL was evaluated by using 30FPS video frames, which means the actual processing speed of target frames should be 5 times slower than the recorded speed. We also tried to use the default 6FPS videos to evaluate LWL, but the performance seriously dropped from 81.4 to 77.4. As to STM, their authors haven't released their evaluation code on YouTube-VOS. Therefore, we mainly compared our speed with CFBI in our submission. Considering CFBI as one of the best high-performance VOS methods, we suppose our comparison of speed is still sufficient.
>
> Notably, we fully optimized the PyTorch implementation of AOT after the submission. The long-term attention was accelerated by more than 1.5X, and the short-term attention was accelerated by more than 20X. The current speeds (on 480p videos) of AOTs are accelerated as listed below.
>
> | Model | AOT-T | AOT-S | AOT-B | AOT-L | CFBI |
> | :-----|:-----:|:-----:|:-----:|:-----:|:-----:|
> | FPS(before) | 37.1 | 18.7 | 12.3 | 8.0 | 5.9|
> | FPS(now) | **39.1** | **29.0** | **22.7** | **18.9** | 5.9 |
>
> The acceleration is more significant when using more LSTT layers. The speedup of AOT-L is more than 2X, leading to a more than 3X speed of CFBI (18.9FPS v.s. 5.9FPS). All the optimized implementations will be publicly available as well.
>
> **Suggestions (minor)**:
>
> **C1**: It would be interesting to evaluate a variant of the method which independently segments each object using the long short-term transformer.
>
> **A1**: Thanks for the suggestion. We will discuss related results in the revision. In fact, we experimented with M=1 and used the soft aggregation to aggregate multiple objects' predictions following the previous common setting [3]. The performance of AOT-S dropped from 80.3% to 78.7%.
>
> We also tried to use the soft aggregation with the default AOT (with 10 identities). In detail, we can predict objects ten by ten for processing the scenarios with more than 10 objects. By doing this, we have successfully applied AOT (with 10 identities) in the application of annotating videos with more than 45 objects. With the auxiliary of AOT, we are collecting a new large-scale video segmentation dataset, whose scenarios can contain more than 45 different objects at most. We believe AOT can learn to associate more identities from this dataset. We will release the dataset for research purposes in the future.
>
> **C2**: What would be the performance drop if random identification vectors are used instead of the learned ones?
>
> **A2**: We have tried to use random identification vectors with orthogonal initialization or Xavier initialization, but the convergence of AOT was very slow, and the IoU score on the training set was always lower than when using learned identities by more than absolute 20% (e.g., 70%: 95%). We will discuss these results in the supplementary material.
>
> **C3**: Did the authors experiment with any other architectures to integrate the identity embedding into the value?
>
> **A3**: In the submission, we tried our best to make AOT as simple as possible since we hope our AOT can ease future research and applications of VOS as much as possible. Thus, as commonly used positional embedding in transformers, we apply identification embedding by simply adding it to features. However, we agree with you that the technical study of other architectures to apply the identification embedding is valuable. For example, we are exploring some architectures which can regularize different identities to be equidistant and thus leads to better robustness with more identities.
>
> **C4**: What is the performance if only the short-term (local) attention is used?
>
> **A4**: According to our experimental results, the performance of AOT-S dropped from 80.3% to 74.9% without long-term attention (similar to the result without short-term attention, 74.3%). Both the long-term guidance and temporal smoothness are important to VOS. Without long-term attention, AOT can not recover objects after occlusion. Without short-term attention, AOT is difficult to be robust to objects' motion and appearance change. This conclusion has also been validated by FEELVOS [6] and CFBI [7].

---

> > ### Comment · Reviewer_tUYE · 2021-08-27
> > **Post rebuttal comments**
> >
> > I thank the authors for providing a detailed rebuttal. The authors answer most of my questions satisfactorily. I will stick with my original rating of "8:  clear accept". I suggest the authors include the additional experiments provided in the rebuttal in the supplementary material or the main paper.

---

### Official Review · Reviewer_kjR8 · 2021-07-16

**Rating:** 7
**Confidence:** 5

**Summary:**

In this paper, the authors study better and efficient way to associate objects in the multi-object video object segmentation. In contrast with previous methods that segment each target object separately, the proposed method matches and decodes multiple objects uniformly and efficiently. For this, an identity bank and a Long Short-Term Transformer is designed. The proposed method AOT become the new state-of-the-art on Youtube-VOS and DAVIS benchmarks.

**Limitations And Societal Impact:**

While future works are discussed in Supplementary Materials, I encourage the authors to include more discussions on limitations and societal impacts.

**Main Review:**

Strength:
- The proposed method present a new paradigm for the semi-supervised VOS deviant from predominant STM-based methods. The proposed method, by design, is efficient when handling video with a large number of objects. e.g., it has constant computational complexity regardless of the number of objects.
- The authors proposed several novel technical components. 1) a new identification mechanism is proposed. An identity bank is employed to assign identities into feature map. With this mechanism, pixel matching become identity-agnostic thus matching results can be shared across targets. 2) a LSTT block is proposed. For my understanding, it is an improved version of memory reading block in STM with transformer style structure and separation of long and short term attention.
- The proposed methods show the state-of-the-art performance on three popular benchmarks.
- The paper includes an extensive ablation analysis.

Weakness:
- I have some concerns on identification mechanism based on identity bank. 1) Scalability. As shown in Table 3 (a), the performance is getting worse with growth of the maximum number of identities. It means that the capacity should be preset to some small number (e.g., 10). In real-world scenario, we can have more than 10 objects and most of the time we don't know how many objects we will need to handle in the future. Have the authors thought about how to scale up without compromising performance? 2) Randomness. Identities are randomly assigned one embedding from the identity bank. How the results are robust against this randomness? It would be undesirable for the result to change with each inference. It would be great to have some analysis on this aspect.

Overall Evaluation:
- The paper present a novel approach for multi-object video object segmentation and the proposed method outperfrom previous state-of-the-arts on several benchmarks.
- Now, I would recommend to accept this paper. I will finalize the score after seeing how authors address my concerns in Weakness.


**Time Spent Reviewing:**

3

---

> ### Author Response · Authors · 2021-08-09
> **Rebuttal**
>
> We appreciate you taking the time to share your comments and suggestions in the review assessment. We are glad that you and all the other reviewers recognize our paper’s novelty and contributions. All your comments are addressed point by point in the following.
>
> **Weaknesses**:
>
> **Q1**: Scalability. In real-world scenarios, we can have more than 10 objects.
>
> **A1**: This is a very valuable consideration, and we have taken this into account in our implementation. In our implementation, AOT (trained with 10 identities) can also process the scenarios with any number (more than 10) of objects. By using soft aggregation, STM [3] predicts objects one by one and aggregates all the predictions together under multi-object scenarios. Following such a simple strategy, AOT (with 10 identities) can predict objects ten by ten for processing the scenarios with more than 10 objects. For example, we can separate 45 objects into 5 object groups (10, 10, 10, 10, and 5 objects), use AOT to process each group, and then aggregate all 45 predicted logits by using soft aggregation.By doing this, we have successfully applied AOT (with 10 identities) in the application of annotating videos with more than 45 objects. With the auxiliary of AOT, we are collecting a new large-scale video segmentation dataset, whose scenarios can contain more than 45 different objects at most. We believe AOT can learn to associate more identities from this dataset. We will release the dataset for research purposes in the future.
>
> **Q2**: Randomness. How are the results robust against this randomness? It would be undesirable for the result to change with each inference.
>
> **A2**: To make sure all the experimental results are reproducible and not random, we didn’t randomly assign identities in the inference stage. When evaluating AOT with N objects, we assigned the first N identities to the objects in their order.
> Besides, the results of AOT were robust, even though we introduced identity randomness during inference. We trained AOT-L for 5 times with different random seeds, and the standard variance of their performance on YouTube-VOS is about 0.1%.
>
> **Limitations And Societal Impact (minor)**:
>
> **Q**: I encourage the authors to include more discussions on limitations and societal impacts.
>
> **A**: As to limitations, we discussed that AOT failed on rare/small objects in Fig. 3 and L305~310, which are also noticed by reviewer HPRW and tUYE. However, we still agree with your valuable suggestion that more discussions on limitations and societal impacts will make our article better. Reviewer Btvt advises us to add privacy concerns about video surveillance systems, and we will discuss them in a new section of border impact.

---

> > ### Comment · Reviewer_kjR8 · 2021-08-31
> > **Post rebuttal comments**
> >
> > I raised two main concerns during the review (scalability and randomness) and the authors addressed both concerns well. I look forward to seeing the experiments with videos with a lot of objects. I will keep my initial rating as 7.

---

### Official Review · Reviewer_Btvt · 2021-07-17

**Rating:** 8
**Confidence:** 4

**Summary:**

This submission addresses the task of semi-supervised video object segmentation (VOS), in which given a set of object mask over a frame, these masks must be propagated to the rest of the frames of the sequence.

The main contribution is proposing a method to treat the multiple objects to be segmented in a single pass, by defining an Identification Embedding that encodes all masks simultaneously. Previous works in VOS used independent embeddings for each object, which required multiple decoding passes, one for each object. With the proposed Identification Embedding and Identification Decoding mechanisms which are trained end-to-end.

The submission also introduced a  Long Short-Term Transformer (LSTT) combines a long-term attention related to the first frame, where the object segments are provided as ground truth, together with a short -term attention related to nearby frames.

The experiments on YouTube-VOS and DAVIS set new state of the art results in terms of accuracy and speed at inference time, especially in the case of multiple object in the video scene.

**Limitations And Societal Impact:**

No, the authors did not include any reference.

The authors may refer to the privacy concerns that video surveillance systems pose, as they may benefit from the presented contributions.

**Main Review:**

## Update on August 27, 2021
After reading the answers of the authors to my concerns, I upgraded my score from "7: Good paper, accept" to "8: Top 50% of accepted NeurIPS papers, clear accept".

## Strengths

S1 Using a single embedding for all the objects allows decoding all the objects in the scene in a single pass, while the related work would require a different decoding step for each segmented object, in such a way that the computational requires will grow with the amount of objects.

S2 The mechanism to develop a trainable scheme for identification embedding and decoding is novel and sounding.

S3 The hierarchical attention mechanism, especially in the case of short-term attention, is convincing in the sense that it attends to nearby temporal frames and nearby spatial pixels frames, in addition to the further frames, attended at a global spatial scale.

S4 The experimental results are careful designed to provide variations of AOT that are adopt the design choice of other works in the state of the art.

S5 The presented results are convincingly faster than the state of the art, together with an incremental gain in accuracy.

S6 The ablation study proves the gains of the main contributions of the paper.

S7 The source code is promised upon acceptance.



## Weaknesses

W1 The concept of multiple object association is not also tested in the very similar problem of multiple object tracking, which should benefit from the same reasoning.

W2 The manuscript uses the adjective "uniformly" (l.35, l.41...) as a contrast to the "independently" (l.32), but it is not clear what "uniformly" refers to in this context. It seems to indicate "simultaneously" because of the reference to the computational requirements and the fact that a single embedding encodes all the objects. The use of the "uniform" term should be better explained or reconsidered.

W3 Is it not clear whether the backbone used by the other works is comparable to the one used in AOT (MobileNet). The authors should discuss whether the gains may come from the different backbones and include it in the table with comparative results.

W4 A discussion regarding the memory footprint of the model and its required memory at inference time is missing. This would also allow a better comparison with the related work, beyond the accuracy and inference times reported.

W5 While the reported frames per second are better than the related work, it is not clear if the results are comparable in terms of hardware. How were the FPS results obtained ? Is it possible that the speed gains are simply due to the use of the distributed 4 Tesla V100 GPU mentioned in l.256 ?

W6 A discussion to this missing related work is missing:

Zhang, D., Zhang, H., Tang, J., Wang, M., Hua, X., & Sun, Q. Feature pyramid transformer. ECCV 2020.



## QUESTIONS

Q1 In Table 3a, did you experiment also with M<10 ? While M=10 is well grounded based on the related work, it is actually one more hyperparameter whose modification may provide some additional gain in performance.

Q2 Similarly to Q1, did you try with larger window sizes ? When reporting results, it is somehow surprising that the next step after the best results ($\lambda=9$) is not included.

## OBSERVATIONS

O1 While CVPR 2021 was celebrated after the NeurIPS 2021 submission deadline, the authors may  **optionally** also consider including some CVPR 2021 papers in the discussion and results on video object segmentation.

Duke, B., Ahmed, A., Wolf, C., Aarabi, P., & Taylor, G. W. [SSTVO: Sparse spatiotemporal transformers for video object segmentation](https://openaccess.thecvf.com/content/CVPR2021/html/Duke_SSTVOS_Sparse_Spatiotemporal_Transformers_for_Video_Object_Segmentation_CVPR_2021_paper.html). CVPR 2021. [[code]](https://github.com/dukebw/SSTVOS)

Wang, Y., Xu, Z., Wang, X., Shen, C., Cheng, B., Shen, H., & Xia, H. [End-to-end video instance segmentation with transformers]((https://openaccess.thecvf.com/content/CVPR2021/html/Wang_End-to-End_Video_Instance_Segmentation_With_Transformers_CVPR_2021_paper.html)). CVPR 2021. [[code]](https://github.com/Epiphqny/VisTR)

Vaswani, Ashish, Prajit Ramachandran, Aravind Srinivas, Niki Parmar, Blake Hechtman, and Jonathon Shlens. ["Scaling Local Self-Attention For Parameter Efficient Visual Backbones."](https://openaccess.thecvf.com/content/CVPR2021/html/Vaswani_Scaling_Local_Self-Attention_for_Parameter_Efficient_Visual_Backbones_CVPR_2021_paper.html) CVPR 2021.

O2 Similarly to O1, the authors may  **optionally** consider citing and discussing the arXivs introducing SwinTransformers and its application to video:

Liu, Z., Lin, Y., Cao, Y., Hu, H., Wei, Y., Zhang, Z., ... & Guo, B. (2021). [Swin transformer: Hierarchical vision transformer using shifted windows](https://arxiv.org/abs/2103.14030). arXiv preprint arXiv:2103.14030.

Liu, Z., Ning, J., Cao, Y., Wei, Y., Zhang, Z., Lin, S., & Hu, H. (2021). [Video Swin Transformer](https://arxiv.org/abs/2106.13230). arXiv preprint arXiv:2106.13230.

O4 l.30: "Above methods" is mentioned twice, it could be rephrased.




**Time Spent Reviewing:**

4

---

> ### Author Response · Authors · 2021-08-09
> **Rebuttal**
>
> We appreciate you taking the time to share your comments and suggestions in the review assessment. We are glad that you and all the other reviewers recognize our paper’s novelty and contributions. All your comments are addressed point by point in the following.
>
> **Weaknesses**:
>
> **Q1**: The concept of multiple object association is not also tested in the very similar problem of multiple object tracking.
>
> **A1**: Thank you for your suggestion. In the supplementary material, we also pointed out that our object association may benefit the tasks requiring multi-object matching, such as multi-object tracking (MOT) and video instance segmentation (VIS). However, this doesn’t mean that the object association can be easily combined with the SOTA methods in these tasks. Different from VOS, the target objects in MOT and VIS are not given and can firstly appear in any frame of the videos. In other words, not only object matching but also object detection is important and necessary for these tasks. However, how to utilize our object association in matching detected objects but keep the solutions still robust to discover new objects during the inference? How to assign an identity to an ambiguous “new target”? These are still open and difficult problems in these tasks. In recent months, we have been working on exploring elegant solutions for these problems with our AOT and have achieved some progress. We will release more technical details in the near future.
>
>
> **Q2**: Why use “uniformly” instead of “simultaneously”?
>
> **A2**: It is great that you noticed this wording detail since we did not use “simultaneously” after careful consideration. Although previous VOS solutions require an individual decoding pass for each object, it is possible to simultaneously conduct all the decoding passes in engineering. For example, all the passes can be distributed to multiple computational threads in parallel and then be completed simultaneously. Thus, we used “uniformly” instead of “simultaneously” to indicate that the processing of multiple objects can be completed within a single pass of AOT. We will add the discussion in the revision.
>
>
> **Q3**: It is not clear whether the backbone used by the other works is comparable to the one used in AOT (MobileNet).
>
> **A3**: All previous SOTA methods use backbones much stronger than MobileNet-V2, such as ResNet-50 (by STM, EGMN, KMN, and LWL) and ResNet-101-DeepLabV3+ (by CFBI). In common, using a stronger backbone always leads to stronger performance in deep-learning-based vision tasks. For example, the DAVIS performance of CFBI will drop from 81.9% to 78.4% by replacing ResNet-101 with weaker MobileNet-V2, as reported in its official repository (https://github.com/z-x-yang/CFBI). Hence, as stated in L232, the use of only a lightweight backbone encoder can further prove that our identification mechanism and LSTT are effective and promising. By the way, we have also tested stronger backbones (such as ResNeSt-101) in AOT-Large and improved the performance by about 1% on YouTube-VOS. We will supply the results with stronger backbones in the supplementary material.
>
> In this work, we aim at making AOT as simple as possible since we hope our AOT can ease future research and applications of VOS as much as possible. Apart from the backbone encoder, our FPN-like decoder network is also much more lightweight than previous works. For example, STM and CFBI used multiple stages of Res-Blocks to construct their decoders.
>
>
> **Q4**: A discussion regarding the memory footprint of the model and its required memory at inference time is missing.
>
> **A4**: Thank you for your suggestion. The parameter numbers and peak inference memory usages (on 480p videos) of AOT variants are listed below in comparison with CFBI.
>
> | Model | AOT-T | AOT-S | AOT-B | AOT-L | CFBI |
> |:----------------- |:--------:|:-----:|:-----:|:-----:|:----:|
> | Param (M) | **5.7** | 7.0 | 8.3 | 8.3 | 66.1 |
> | Mem (G) | **0.56** | 0.58 | 0.61 | 3.6 | 4.7 |
> | J&F (YouTube-VOS) | 80.2 | 82.6 | 83.2 | **83.7** | 81.4 |
>
> As shown in the table, AOT-T/S/B takes only less than 13% of CFBI’s parameter number and memory usage. The memory usage of AOT-L can increase from 0.61G to 3.6G, along with the updation of long-term memory during inference. However, the peak memory usage of AOT-L is still less than CFBI (3.6 v.s. 4.7). In summary, AOT is significantly more lightweight than CFBI. We will supply this comparison into the revision.
>
>
> **Q5**: How were the FPS results obtained?
>
> **A5**: For the sake of fairness, we recorded the FPS results of AOT variants under the same kind of device used by CFBI (GPU: 1 Tesla V100). Notably, we fully optimized the PyTorch implementation of AOT after the submission. The long-term attention was accelerated by more than 1.5X, and the short-term attention was accelerated by more than 20X. The current speeds (on 480p videos) of AOTs are accelerated as listed below.
>
> | Model | AOT-T | AOT-S | AOT-B | AOT-L | CFBI |
> | :-----|:-----:|:-----:|:-----:|:-----:|:-----:|
> | FPS(before) | 37.1 | 18.7 | 12.3 | 8.0 | 5.9|
> | FPS(now) | **39.1** | **29.0** | **22.7** | **18.9** | 5.9 |
>
> The acceleration is more significant when using more LSTT layers. The speedup of AOT-L is more than 2X, leading to a more than 3X speed of CFBI (18.9FPS v.s. 5.9FPS). All the optimized implementations will be publicly available as well.
>
>
> **Q6**: Missing related work.
>
> **A6**: Thank you for your suggestion. We will discuss the mentioned article in the related works of vision transformers.
>
>
> **Questions (minor)**:
>
> **Q1**: Did you also experiment with M<10?
>
> **A1**: Yes. We did an experiment with M=1 and used the soft aggregation to aggregate multiple objects’ predictions following the previous typical setting [3]. The performance seriously dropped from 80.3% to 78.7%. As to M numbers between 1 and 10, we didn’t study them since we hope M to be as large as possible to cover all the objects. There are some scenarios in YouTube-VOS that contain up to 10 objects, and M has to be at least 10 to cover all of them.
>
>
> **Q2**: Did you try with larger window sizes (> 7)?
>
> **A2**: Yes. We did an experiment with a window size of 9, and the improvement over the window size of 7 was less than absolute 0.2%. Moreover, the window size of 9 increased the inference memory usage of AOT-S by about 27% and decreased the inference speed by about 10%. Since the motion offsets between two continuous video frames commonly have an upper bound, we suppose the effective window size of local attention has a corresponding upper bound, which should be close to 7. In other words, using a window size larger than 7 can not efficiently improve the performance of local attention further. We will supply related discussion in the revision.
>
>
> **Observations (minor)**:
>
> **Q**: Suggestions of references. Typos.
>
> **A**: Thank you for your careful review and valuable suggestions. We will discuss more with the mentioned VOS/VIS methods and vision transformers. All the typos will be corrected in the revision.
>
>
> **Limitations And Societal Impact (minor)**:
>
> **Q**: The authors may refer to the privacy concerns that video surveillance systems pose, as they may benefit from the presented contributions.
>
> **A**: Thanks for such a thoughtful suggestion. We will add the privacy concerns into the border impact.

---

### Official Review · Reviewer_HPRW · 2021-07-17

**Rating:** 8
**Confidence:** 3

**Summary:**

This work proposes two novel techniques, an identity matching and a long-short term transformer block. Their first contribution, identity matching, enables them to handle multiple object scenarios as fast as single object ones and process all objects simultaneously. Their second contribution, enables them to handle occlusion better while keeping boundaries accurate. Overall they achieve sota results on 3 main semi supervised video object segmentation tasks.

**Limitations And Societal Impact:**

Authors have noted that this method fails on rare/small objects. They also claim since this is a standard benchmark their work does not deviate or add any negative impact.

**Main Review:**

This work proposes to initialize M identity vectors (M is set to 10 in practice which is the maximum number of objects). Then using a random permutation matrix N  (number of present objects) of them are assigned to the current video. They augment the value vector of the memory with the identity embedding by adding them together. With this strategy they are able to track multiple objects simultaneously with a matrix multiplication and addition. This is a novel and interesting contribution which enables them to get a very high FPS while achieving sota. In extreme with their tiny variant they are able to achieve about 40 FPS processing which is impressive given their accuracy.

Their other main contribution is having two attention heads, one of them looks at the local spatial positions in the previous frame (7x7 very local!) and the other one looks at the whole image but from previous frames. The long attention either only looks at the first frame or at every 5 frame in the memory constructing their best large model.

Experimentally they show sota results on both DAVIS and YoutubeVOS which are the two main semi supervised benchmarks and they outperform all the previous works significantly. Impressively while maintaining a better FPS too.

Both of their contributions are significant in terms of performance while not adding too much complexity or computational overhead.

They also perform an ablation study on all the interesting factors of their model, such as number of LSTT blocks, number of short term history frames and window size.

Their performance drops with more identity vectors than necessary (such as 20 or 30). An interesting visualization in appendix shows that the identity vectors start getting correlated. Their hypothesis is it's harder to optimize for 30 disjoint vectors in a 250D space. It would be interesting if this can be resolved with a proper regularizer on the identity vectors such as orthogonality enforcer.

**Time Spent Reviewing:**

3

---

> ### Author Response · Authors · 2021-08-09
> **Rebuttal**
>
> We appreciate you taking the time to share your comments and suggestions in the review assessment. We are glad that you and all the other reviewers recognize our paper's novelty and contributions. All your comments are addressed point by point in the following.
>
> **Suggestions (minor)**:
>
> **Q**: An interesting visualization in the Appendix shows that the identity vectors start getting correlated (with more identity vectors). Their hypothesis is it's harder to optimize for 30 disjoint vectors in a 256D space.  It would be interesting if this (the performance drops with more identity vectors) can be resolved with a proper regularizer on the identity vectors such as orthogonality enforcer.
>
> **A**: Thank you for the valuable suggestion. We also believe that designing a regularizer (such as the mentioned orthogonality enforcer) is possible to avoid identity vectors getting correlated and thus relieve the performance drops from using more identities. Therefore, we are exploring related solutions.
>
> To relieve the performance drops, we are also working on collecting a new large-scale video segmentation dataset, whose scenarios can contain more than 45 different objects at most. As mentioned in Appendix, another hypothesis for the performance loss is that no training video contains enough objects to be assigned so many identities (when using more than 10 identities). Thus the network cannot learn to associate all the identities simultaneously. Accordingly, we suppose AOT can learn to associate more identities from our new dataset with more objects. We will release our dataset for research purposes in the future.
>
> Apart from increasing the number of identity vectors, AOTs trained with M identities can also be applied to scenarios with more than M objects to avoid using more identities. For example, using soft aggregation [3], AOT (with 10 identities) can predict objects ten by ten for processing the scenarios with more than 10 objects. By doing this, we have successfully applied AOT (trained with 10 identities) to assist the annotation of videos (in our new dataset) with more than 45 objects.
>
> **Limitations And Societal Impact  (minor)**:
>
> **Q**: Authors have noted that this method fails on rare/small objects.
>
> **A**: Rare/small objects are common challenging cases for all segmentation-related tasks. In the VOS area, this problem also exists in STM [3] and its following works [4,5]. These methods do matching processes on feature maps with a low resolution (e.g., 1/16 of the input images), leading to worse performance on tiny objects. For the sake of fairness, AOT follows this common setting to perform matching processes. We suppose that using feature maps with higher resolutions like CFBI [7] can relieve the problem. However, directly increasing the feature resolution is not novel and will lead to much more computation. Thus, we are exploring some more efficient ways of solving this.

---

### Decision · Program_Chairs · 2021-09-27

**Decision:**

Accept (Poster)

**Comment:**

The rebuttal addressed all of the reviewers concerns, and all reviewers recommend acceptance. The AC agrees with this recommendation.